# REVISITING CONVOLUTION ARCHITECTURE IN THE REALM OF DNA FOUNDATION MODELS

**Yu Bo**[1][*], **Weian Mao**[1,2][*], **Yanjun Shao**[3], **Weiqiang Bai**[4], **Peng Ye**[4]
**Xinzhu Ma**[4], **Junbo Zhao**[1], **Hao Chen**[1], **Chunhua Shen**[6,1,5]

[1] Zhejiang University    [2] MIT, USA    [3] Yale University, USA
[4] Shanghai AI Lab    [5] Ant Group    [6] Zhejiang University of Technology

## ABSTRACT

In recent years, a variety of methods based on Transformer and state space model (SSM) architectures have been proposed, advancing foundational DNA language models. However, there is a lack of comparison between these recent approaches and the classical architecture—convolutional networks (CNNs)—on foundation model benchmarks. This raises the question: *Are CNNs truly being surpassed by these recent approaches based on transformer and SSM architectures?* In this paper, we develop a simple but well-designed CNN-based method, termed **ConvNova**. ConvNova identifies and proposes three effective designs: 1) dilated convolutions, 2) gated convolutions, and 3) a dual-branch framework for gating mechanisms. Through extensive empirical experiments, we demonstrate that ConvNova significantly outperforms recent methods on more than half of the tasks across several foundation model benchmarks. For example, in histone-related tasks, ConvNova exceeds the second-best method by an average of 5.8%, while generally utilizing fewer parameters and allowing faster computation. In addition, the experiments observed findings that may be related to biological characteristics. This indicates that CNNs are still a strong competitor compared to Transformers and SSMs. We anticipate that this work will spark renewed interest in CNN-based methods for DNA foundation models. Code is available at: https://github.com/aim-uofa/ConvNova

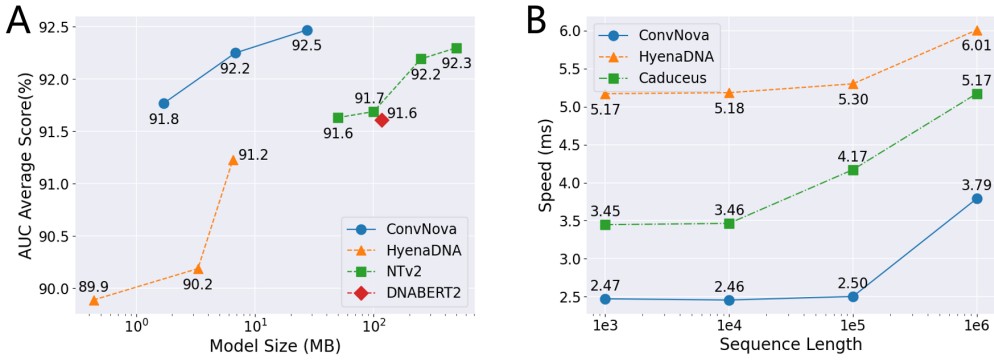

Figure 1: A) The trade-off between model size and accuracy (AUC score) of various methods. ConvNova achieves the current SoTA performance. B) The trade-off between input sequence length and runtime of different methods. ConvNova exhibits clear superiority over HyenaDNA and Caduceus. All models are around 7M parameters, tested on A100 80GB with batch size 1.

---

[*]YB and WM contributed equally. Work was done when WM was visiting Zhejiang University. CS and HC are the corresponding authors.

# 1 INTRODUCTION

Consistently at the forefront of scientific research, DNA has catalyzed significant advancements across a multitude of fields, including aging research, synthetic biology, and disease treatment (Chucair-Elliott et al., 2024; Moe-Behrens et al., 2013). In contrast, unlike NLP, comprehending DNA poses a formidable challenge, owing to the myriad undiscovered biological principles underlying its structure and function. Fortunately, the rapid progress of genome language modeling has showcased its dominance in numerous subsequent applications. These include the prediction of promoters (Zhang et al., 2022), gene expression (Avsec et al., 2021), DNA methylation (Jin et al., 2022), analysis of chromatin state (Lee et al., 2022), prediction of promoter-enhancer interactions (Chen et al., 2022), TF-DNA binding prediction (Wang et al., 2022), variant effect prediction (Rozowsky et al., 2023), and gene network prediction (Theodoris et al., 2023).

Versatile sequence models have been established as foundational models for genetics, typically falling into two categories: Transformer-based methods (*e.g.*, DNABERT (Ji et al., 2021), DNABERT-2 (Zhou et al., 2023), NucleotideTransformer (Dalla-Torre et al., 2023)) and SSM-inspired methods (*e.g.*, HyenaDNA (Nguyen et al., 2024), Caduceus (Schiff et al., 2024)). These approaches have made significant progress in genomic language modeling tasks and are now widely adopted. Prior to these developments, however, convolutional neural networks (CNNs) were the dominant technique for DNA modeling in bioinformatics, giving rise to notable models such as Basset (Kelley et al., 2016), Bassenji (Kelley et al., 2018), and LegNet (Penzar et al., 2023). Despite their success, CNN architectures have not been systematically compared to the newer DNA foundation models in recent research. Comparisons between CNNs and modern approaches have been limited to specific domains, such as NTv2 (Dalla-Torre et al., 2023) and Caduceus (Schiff et al., 2024). In these cases, only a few domain-specific CNN models, like Spliceator (Scalzitti et al., 2021) and some benchmark baselines, have been evaluated.

In this landscape, we rethink and re-evaluate *whether CNN methods are indeed less effective than the current leading paradigms, including Transformer-based methods and SSM-inspired methods*, which have been the primary focus of recent research in DNA foundation models. The motivation behind this is straightforward, as CNNs still hold advantages over existing methods: 1) The input lengths for downstream tasks on DNA vary greatly (ranging from tens to thousands). Transformers are not robust to sequence length variations (Press et al., 2021), and their performance can be affected. 2) Transformers have a computational complexity of $O(n^2)$, higher than CNN methods. Additionally, to avoid overly long sequences, transformers also require tokenizers, which affect translational invariance (Zhou et al., 2023). 3) Mamba (Gu and Dao, 2023) is better suited for tasks involving long sequences with autoregressive properties (Yu and Wang, 2024), while DNA sequences typically do not exhibit autoregressive characteristics. 4) CNNs possess a local inductive bias, which can provide better data utilization efficiency and are more friendly towards tasks with small training data volumes.

Based on this motivation, we propose a simple yet well-designed CNN method, named ConvNova. Extensive empirical experiments demonstrate that ConvNova outperforms recent Transformer and SSM-inspired methods on more than half of the evaluated tasks, indicating that CNNs continue to demonstrate superiority. Through an analysis of the design of ConvNova, we identify and propose three key designs that enable CNNs to achieve superior performance: 1) *dilated convolution* (Yu and Koltun, 2015), 2) *gated convolution* (Yu et al., 2019), and 3) *a dual-branch framework for the gated convolution*. Specifically, we found that for DNA tasks, increasing the receptive field of the CNN through downsampling (U-Net (Ronneberger et al., 2015) style) can severely degrade CNN performance. This stands in contrast to observations in other fields, such as computer vision. However, dilated convolution can circumvent this issue by enlarging the receptive field without the need for downsampling. Additionally, gated convolution can significantly enhance CNN performance. We hypothesize that this is because DNA may contain a substantial amount of irrelevant segments, and the gating mechanism can suppress this information. Lastly, we discovered that splitting the network pathways into two branches, with one branch exclusively providing gating signals to the other, can improve network performance. For detailed designs, please refer to §3 and §4.

Comprehensive experiments demonstrate the superiority of ConvNova. On the NT benchmarks, ConvNova achieves SoTA performance on 12 out of 18 datasets. Notably, in the H3K4me3 task, it outperforms the second-best method by 10.5%, and in the histone-related tasks, it surpasses the

second-best method by an average of 5.8%. In the gene finding benchmark (long sequences modeling), ConvNova significantly surpasses HyenaDNA (Nguyen et al., 2024) (55% vs. 35%). Compared to the transformer model with much larger parameters (7.4M vs. 336M), ConvNova still achieves better performance (55% vs. 52%).

By adjusting the receptive field size of ConvNova, specifically through controlling the dilation stride, we reveal that the performance of the vast majority of tasks improves with the increase of the receptive field of the CNN. However, we find that some specific tasks, including H3K4me2, H3K4me3, and H3K14ac, exhibit the opposite phenomenon. Existing wet lab experimental data indicate that some of these tasks exhibit pronounced localized characteristics; therefore, this phenomenon may be caused by biological characteristics. For detailed findings and analysis, please refer to §5.

In summary, the main contributions of our work are as follows:

- We reexamine the convolutional paradigm for DNA foundation models. Analyses and extensive experimentation demonstrate the continued competitiveness of convolutional neural networks (CNNs) for downstream tasks. This may prompt the community to reconsider CNNs for such tasks.

- We propose ConvNova and identify three key designs for genomic language modeling: dilated convolution, gated convolution, and a dual-branch design. With these designs, ConvNova achieves superior performance on numerous benchmarks while maintaining fewer parameters.

- ConvNova outperforms recent methods on over half of the tasks across various DNA foundation model benchmarks, typically using fewer parameters and achieving faster performance.

## 2 PRELIMINARIES AND RELATED WORK

### 2.1 TRANSFORMER-BASED DNA MODELS

Transformers have become a dominant force across various fields in the deep learning community, and recent research has begun to explore their potential within the realm of genomics. DNABERT (Ji et al., 2021) was the first to establish the transformer as a foundational model for DNA, utilizing k-mer tokenization. Nucleotide Transformer (NT) (Dalla-Torre et al., 2023) scaled up the transformer model (with sizes of 500M and 2.5B) and investigated the impact of different pretraining datasets (HG38[1] and Multispecies[2]) on 18 downstream tasks. Their experiments demonstrate that multispecies data and larger models generally yield superior performance. DNABERT-2 (Zhou et al., 2023), also pretrained on multispecies data, tackled several problems encountered with DNABERT. They further explored the Byte Pair Encoding (BPE) tokenization and ALiBi positional embedding as effective strategies for modeling DNA with transformer models. Additionally, they utilized FlashAttention to accelerate the processing. Meanwhile, BigBird (Zaheer et al., 2020), despite not being specifically designed, had also been applied in genomics.

While adept at capturing long-range dependencies, transformer-based models in genomics face challenges compared to CNNs. CNNs possess an inductive bias towards local, translation-invariant features (Battaglia et al., 2018), aligning well with many genomic patterns. In contrast, transformers lack this inherent bias for local structure modeling. Additionally, the attention layer's $O(n^2)$ complexity poses computational challenges, especially for long DNA sequences.

### 2.2 SSM-INSPIRED DNA MODELS

Another approach that has gained traction involves the use of State Space Models (SSMs). HyenaDNA (Nguyen et al., 2024) is a decoder-only, sequence-to-sequence model that incorporates a modified convolution operator (Hyena (Poli et al., 2023)). This innovative approach enables the expansion of input sequences of the model to 1M-size nucleotides and significantly outperforms transformer-based models in a variety of tasks with a model size that is 300 times smaller. Additionally, Mamba (Gu and

---

[1]https://www.ncbi.nlm.nih.gov/assembly/GCF_000001405.26
[2]based on genomes at https://www.ncbi.nlm.nih.gov/assembly/GCF_000001405.26; more details can refer to NT's original paper.

Dao, 2023), a state-space-model inspired architecture, has demonstrated promising, albeit preliminary, experiments on DNA sequences. Caduceus (Schiff et al., 2024) refines the Mamba Block into an RC-equivariant architecture. By taking into account the reverse complementary strand, Caduceus surpasses both HyenaDNA and transformer-based models in downstream tasks, marking a significant advancement in the field.

### 2.3 CNN-BASED DNA MODELS

Convolutional Neural Network (CNN)–based DNA models are powerful deep learning tools designed to analyze and interpret DNA sequences by automatically extracting significant motifs and patterns. Early models like DeepBind (Alipanahi et al., 2015) predicted the sequence specificities of DNA and RNA-binding proteins, while DeepSEA (Zhou and Troyanskaya, 2015) used CNNs to predict the effects of noncoding variants. Basset (Kelley et al., 2016) advanced the field by learning the regulatory code of the accessible genome using deep CNNs.

Recent advancements have further enhanced the capabilities of CNN-based genomic models. Enformer (Avsec et al., 2021) integrates convolutional layers with attention mechanisms to predict gene expression from DNA sequences, effectively capturing short- and long-range regulatory interactions. Borzoi (Linder et al., 2023) builds upon the Basenji (Kelley et al., 2018) family by incorporating dilated convolutions and skip connections, improving performance on various genomic prediction tasks. LegNet (Penzar et al., 2023) employs a convolutional architecture for predicting gene expression and single-nucleotide variant effects in regulatory regions, achieving first place in the DREAM 2022 challenge.

These models demonstrate the CNNs' strengths, capturing local sequence features and motifs crucial for understanding genetic information. However, despite their success, they have not been incorporated into the broader consideration of DNA foundation models.

## 3 METHOD

In this section, we first present our overall network architecture (§3.1), followed by an introduction to the core component of this architecture - the Gated Convolution Block (§3.2). Subsequently, we delve into the selection of convolutional network methods, specifically whether to use dilation or down-sampling (§4.4). Finally, we discuss our pretraining approach and downstream usage (§3.3).

### 3.1 CONVNOVA

Figure 2 shows the overall illustration of ConvNova. DNA sequences are mapped to the hidden space through a convolution operation, followed by $N$ gated convolution blocks (GCBs), and finally processed through a multilayer perceptron (MLP). Different output heads are used in pretraining and downstream tasks. Each GCB utilizes dilated convolution to increase the receptive field and aggregate the features through a dual-branch structure.

### 3.2 GATED CONVOLUTION BLOCK

Yu et al. (2019) introduce gated convolution as a solution to a problem inherent to standard convolution, which indiscriminately treats all input pixels as valid. This method enhances partial convolution by offering a learnable dynamic feature selection mechanism for each channel at every spatial location throughout all layers. This mechanism allows for more nuanced and effective feature selection, improving the performance of convolutional neural networks.

With the motivation to effectively retain and forget information, we propose Gated Convolutional Blocks (GCBs). The dual-branch structure of these blocks is designed to facilitate independent feature extraction, thereby promoting complementary representation learning. Let's define $\mathbf{A} \in \mathbb{R}^{l \times d}$ as the feature extracted by the left branch and $\mathbf{B} \in \mathbb{R}^{l \times d}$ as the right branch feature.

We first process the left branch feature $\mathbf{A}$ through a LayerNorm layer and then pass it through a convolution layer, followed by the GELU activation function to obtain the intermediate features $\mathbf{h}$ of the current layer. In the other path, the right branch features $\mathbf{B}$ are processed through other

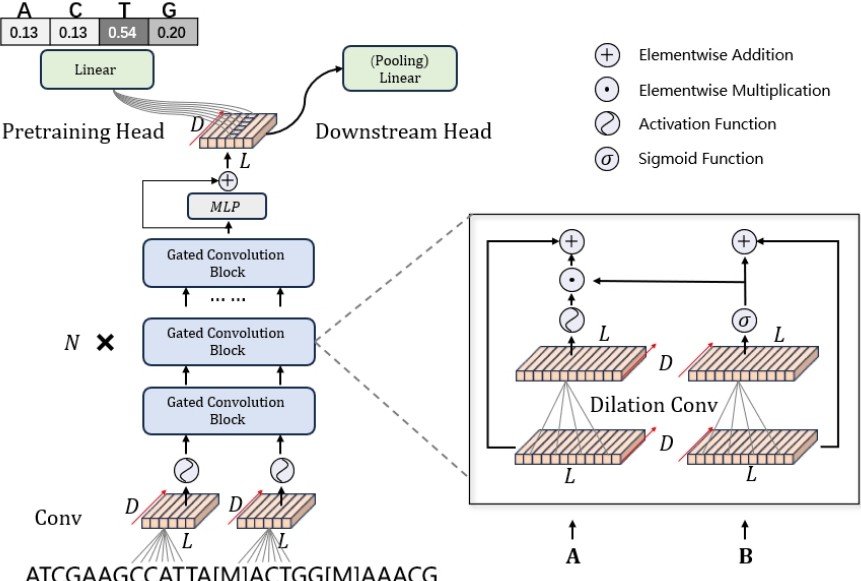

Figure 2: **ConvNova architecture.** The ConvNova model processes double stands as inputs. The sequences are initially subjected to one-hot encoding and subsequently pass through a convolution layer. The processed data then enters a series of Gated Convolution Blocks (GCBs), the specifics of which are elaborated in §3.2. The output from the GCBs is then fed into an MLP. The final stage of the framework bifurcates into two distinct heads: the downstream head and the pretraining head.

LayerNorm and convolution layers that do not share weight with the left branch, followed by the sigmoid activation function to obtain the intermediate features $\mathbf{g}$.

Next, we update the left branch features $\mathbf{A}$ by multiplying the intermediate features $\mathbf{g}$ with the intermediate features $\mathbf{h}$. Simultaneously, the right branch features $\mathbf{g}$ are used to update the right branch features $\mathbf{B}$ themselves. Additionally, the kernel size for all convolutional layers is consistently set to 9. This entire process, which integrates both branches, is described by Eq.1 and is illustrated in the Gated Convolution Block of Figure 2:

$$
\begin{aligned}
\mathbf{h} &= \text{GELU}(\text{convolution}(\mathbf{A})) \\
\mathbf{g} &= \text{sigmoid}(\text{convolution}((\mathbf{B})) \\
\mathbf{A} &= \mathbf{A} + \mathbf{h} \odot \mathbf{g} \\
\mathbf{B} &= \mathbf{B} + \mathbf{g}
\end{aligned}
\tag{1}
$$

A stage is composed of several GCBs, with the dilation rate for each Block set to $[1, 1, d, d^2, d^3, ...]$. In different experiment, we use different stages. By default, 1 stage uses 5 GCBs. You can find the information in Table 9.

### 3.3 MLM PRETRAINING AND DOWNSTREAM USAGE

**MLM Pretraining**   We use the bidirectional masked language model (MLM) pretraining method and have observed significant performance improvement in downstream tasks.

In this work, 10% of the nucleotides in a primary DNA strand are masked, and the model is trained to predict the type of the masked tokens. Therefore no extra labels are required. The objective of this pretraining is to infer the type of nucleotides that have been masked by utilizing the surrounding nucleotides. The optimization objective is to minimize the cross-entropy loss. The pretraining data is HG38, same as HyenaDNA (Nguyen et al., 2024).

**Downstream Usage**   Upon completing the pretraining phase for 400 epochs, we proceed to finetune ConvNova on the downstream tasks, keeping the entire model adjustable. We employ the pooling

Table 1: **NT Benchmark results.** MCC/F1-score is reported for pretrained NTv2, HyenaDNA, DNABERT-2, Caduceus-Ph, and ConvNova. The best values per task are bold, and the second-best are underlined. ± indicates the error range across five random seeds.

| | NTv2 (500M) | HyenaDNA (1.6M) | DNABERT-2 (117M) | Caduceus-Ph (1.9M) | ConvNova (1.7M) |
|---|---|---|---|---|---|
| *Histone* | | | | | |
| H3 | 78.17 ±2.54 | 78.14 ±1.70 | 79.31 ±0.68 | 80.48 ±1.04 | **81.50** ±0.80 |
| H3K4me1 | 51.64 ±1.12 | 44.52 ±2.59 | 48.34 ±4.63 | 52.83 ±0.96 | **56.60** ±1.01 |
| H3K4me2 | 37.24 ±2.25 | 42.68 ±2.66 | 43.02 ±2.92 | 49.88 ±2.65 | **57.45** ±2.27 |
| H3K4me3 | 50.30 ±1.77 | 50.41 ±3.15 | 45.43 ±3.33 | 56.72 ±2.58 | **67.15** ±0.93 |
| H3K9ac | 61.05 ±1.40 | 58.50 ±1.75 | 60.04 ±1.27 | 63.27 ±2.29 | **68.10** ±1.91 |
| H3K14ac | 57.22 ±2.19 | 56.71 ±2.40 | 54.49 ±4.99 | 60.84 ±2.94 | **70.71** ±2.32 |
| H3K36me3 | 60.50 ±1.75 | 59.92 ±1.06 | 57.58 ±2.38 | 61.12 ±1.44 | **68.31** ±1.19 |
| H3K79me3 | 65.78 ±2.34 | 66.25 ±3.65 | 64.38 ±0.48 | 67.17 ±2.03 | **72.08** ±1.23 |
| H4 | 79.87 ±1.34 | 78.15 ±1.58 | 78.18 ±0.98 | 80.10 ±1.00 | **81.12** ±0.93 |
| H4ac | 55.22 ±2.20 | 54.15 ±2.96 | 51.80 ±0.10 | 59.26 ±3.67 | **66.10** ±1.20 |
| *Regulatory* | | | | | |
| Enhancer | 54.51 ±1.94 | 53.13 ±4.52 | 52.50 ±1.44 | 55.20 ±2.56 | **57.60** ±2.52 |
| Enhancer Types | 43.36 ±1.75 | 48.16 ±2.48 | 44.32 ±1.18 | 47.17 ±2.85 | **49.75** ±2.82 |
| Promoter All | **96.82** ±0.47 | 95.57 ±0.18 | 96.23 ±0.17 | 96.65 ±0.16 | **96.82** ±0.22 |
| Promoter non-TATA | **97.45** ±0.69 | 95.86 ±0.37 | 97.17 ±0.17 | 96.31 ±0.50 | 96.76 ±0.21 |
| Promoter TATA | 96.53 ±0.81 | 95.88 ±0.53 | **96.99** ±0.49 | 96.21 ±0.81 | 96.34 ±0.38 |
| *Splice sites* | | | | | |
| Splice Acceptor | **97.99** ±0.66 | 96.98 ±0.49 | 97.49 ±0.36 | 94.21 ±0.37 | 96.23 ±0.41 |
| Splice Donor | **98.50** ±0.43 | 95.27 ±1.07 | 94.33 ±0.27 | 94.69 ±0.67 | 96.62 ±0.61 |
| Splice All | **98.15** ±1.01 | 94.05 ±1.08 | 93.75 ±1.25 | 92.87 ±1.73 | 96.33 ±0.31 |

method of HyenaDNA to get an embedding for each DNA sequence and utilize a linear mapping to the output label dimension to serve as the finetuned head. In cases where it's not significantly indicated, the setting for our downstream tasks remains consistent with that of HyenaDNA (Nguyen et al., 2024).

## 4 EXPERIMENTS

Some other studies have demonstrated that adopting multispecies data for pretraining can enhance downstream task performance. However, in this work, all pretraining tasks are performed on the human reference genome[3] to align with the previous work (HyenaDNA and Caduceus). In addition, we employ a tokenization scheme at the character or base pair level, which has been proven to function well in SSM-inspired models. We also show the speed comparison between ConvNova and other models along sequence length ($10^3$ to $10^6$). At a sequence length of $10^6$, ConvNova is 1.35 times faster than its fastest counterpart–HyenaDNA, as shown in Figure 1. Please refer to A.1 for additional details on the pretraining dataset and recipes.

### 4.1 SHORT RANGE

#### 4.1.1 NUCLEOTIDE TRANSFORMER BENCHMARK

We start the evaluation with the recently proposed Nucleotide Benchmark (Dalla-Torre et al., 2023). These datasets encompass three types of tasks: histone marker prediction, regulatory annotation prediction, and splice site annotation prediction.

To assess performance, we follow the methodology outlined in (Nguyen et al., 2024), applying different metrics depending on the task: Matthews Correlation Coefficient (MCC) for histone marker

---

[3]https://www.ncbi.nlm.nih.gov/assembly/GCF_000001405.26

tasks and various enhancer-related tasks, F1 score for various promoter-related tasks and splice site annotation tasks. The baselines consist of NT-v2(500M), DNABERT-2, similar-sized HyenaDNA, and Caduceus-ph. Because in (Schiff et al., 2024), Caduceus-ph outperforms Caduceus-ps in almost all functions except promoter TATA and H3K36me3, we only choose Caduceus-ph as our baseline. For HyenaDNA and ConvNova, we adopt the pooling methodology from HyenaDNA (Nguyen et al., 2024) and implement a linear layer to derive the logits.

Furthermore, we recognize the instability inherent in the datasets, where different random seeds may result in variations of up to 5 points. Therefore, we provide the mean and standard deviation (±) calculated from 10 random seeds.

The results of this benchmark suite are showcased in Table 1, demonstrating the competitive performance of ConvNova. Notably, ConvNova outperforms Caduceus-ph, the second-best model, by 10 points despite having fewer parameters. Across nearly all tasks, ConvNova exhibits superior performance compared to a similarly sized HyenaDNA model, except for the splice sites acceptor. Additionally, ConvNova surpasses the transformer-based DNABERT-2 in 15 out of 18 tasks and NT-v2 in 12 out of 18 tasks despite having significantly fewer parameters. For more details, see A.2.1.

Furthermore, we have also included previously popular supervised CNNs, LegNet (Penzar et al., 2023) and Basenji (Kelley et al., 2018), in this benchmark. For more details, please refer to A.4.

### 4.1.2 GENOMIC BENCHMARK

Table 2: **Genomic Benchmark results.** Top-1 accuracy (↑) is reported for pretrained HyenaDNA, Caduceus-Ph, ConvNova, and the original CNN baseline (trained from scratch). The best values are in bold, and the second-best is underlined. ± indicates the error range across five random seeds.

| Task | CNN (264K) | HyenaDNA (436K) | Caduceus-Ph (470K) | ConvNova (386K) |
|---|---|---|---|---|
| ***Enhancers*** | | | | |
| Mouse Enhancers | 0.730 ±0.032 | 0.779 ±0.013 | 0.754 ±0.074 | **0.784** ±0.009 |
| Human Enhancers Cohn | 0.702 ±0.021 | 0.718 ±0.008 | **0.747** ±0.004 | 0.743 ±0.005 |
| Human Enhancer Ensembl | 0.744 ±0.122 | 0.832 ±0.006 | 0.893 ±0.008 | **0.900** ±0.004 |
| ***Species Classification*** | | | | |
| Coding vs. Intergenomic | 0.892 ±0.008 | 0.904 ±0.008 | 0.915 ±0.003 | **0.943** ±0.001 |
| Human vs. Worm | 0.942 ±0.002 | 0.961 ±0.002 | **0.973** ±0.001 | 0.967 ±0.002 |
| ***Regulatory Elements*** | | | | |
| Human Regulatory | 0.872 ±0.005 | 0.862 ±0.004 | 0.872 ±0.011 | **0.873** ±0.002 |
| Human Non-TATA Promoters | 0.861 ±0.009 | 0.887 ±0.005 | 0.946 ±0.007 | **0.951** ±0.003 |
| Human OCR Ensembl | 0.698 ±0.013 | 0.744 ±0.019 | **0.828** ±0.006 | 0.793 ±0.004 |

Next, we conduct a comprehensive evaluation using eight datasets introduced by GenomicsBenchmarks (Grešová et al., 2023) with eight regulatory element classification tasks with sequence lengths spanning from 200 to approximately 2000bps. Our baselines comprise HyenaDNA, Caduceus-ph, and the original CNN model described in (Grešová et al., 2023). We conduct experiments using five distinct random seeds and report the mean and the disparity between the maximum/minimum and the mean results.

As shown in Table 2, ConvNova models outperform baselines in 5 tasks. In other tasks, ConvNova is competitive with the best models. Additionally, we also compared the performance of supervised CNNs, LegNet and Basenji, in this benchmark. See A.2.4 and A.4 for additional experiment details.

### 4.2 LONG RANGE

### 4.2.1 BEND GENE FINDING

Gene finding (Marin et al., 2023) is a multiclass classification task aimed at predicting the nucleotide types within DNA sequences (exons, introns, donors, acceptors, noncoding regions). This task

Table 3: **Gene Finding results.** MCC-score is reported for ConvNova and baseline models. The best value is in bold.

| GeneFinding | NT-H (500M) | DNABERT-2 (117M) | GENA-LM (336M) | HyenaDNA (6.5M) | ConvNova (7.4M) |
|---|---|---|---|---|---|
| | 0.41 | 0.43 | 0.52 | 0.35 | **0.55** |

aids in gene annotation and coding sequence identification. The dataset, derived from GENCODE annotations, includes labels $\{E_F, D_F, I_F, A_F, E_R, D_R, I_R, A_R, NC\}$.

The task employs the GENCODE dataset (Harrow et al., 2012), comprising 5,976 DNA sequences spanning 1,433 to 14,000 bps. The objective is to classify each nucleotide based on its gene structure context, using local signals to predict exon-intron boundaries and long-range dependencies for accurate annotation. As shown in Table 3, our ConvNova model achieved the highest MCC of 0.55 on the gene-finding task, outperforming NT-H (0.41), DNABERT-2 (0.43), GENA-LM (0.52), and HyenaDNA (0.35). This demonstrates ConvNova's ability to leverage local and long-range dependencies for improved gene annotation.

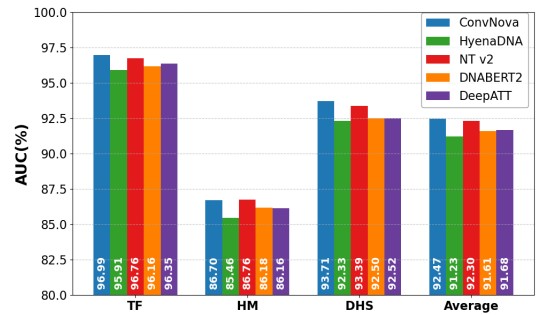

Figure 3: **Chromatin Profile Prediction results.** AUC Score (↑) is reported for ConvNova, HyenaDNA, NT v2, DNABERT-2, and DeepATT performance on transcription factors (TF), DNase I hypersensitive sites (DHS), histone modifications (HM), and average score.

### 4.2.2 CHROMATIN PROFILE PREDICTION

This classification task (Zhou and Troyanskaya, 2015) focuses on predicting chromatin profiles and epigenetic markers from DNA sequences, which is crucial for quantifying the functional effects of non-coding variants. The dataset comprises 919 chromatin features, including transcription factor (TF) binding profiles, DNase I hypersensitive sites (DHS), and histone mark (HM) profiles. For a given DNA sequence of length 1,000 bps, the task involves jointly predicting 919 binary classes corresponding to the chromatin profile of a central region of the sequence. We fine-tune our pretrained ConvNova models for 10 epochs, along with all baseline models, which range from 436K (tiny HyenaDNA) to 500M (NT-v2). Notably, ConvNova outperforms all baseline models, including DeepATT (Li et al., 2021), which is a previously state-of-the-art supervised model for this task, as illustrated in Figure 3.

Moreover, we provide compelling evidence of ConvNova's scalability when ample data is available. As we augment the model's parameter count from 6.8M to 51.1M, ConvNova shows a discernible improvement in the performance capabilities exhibited. See Appendix A.2.2 for experiment details and comparing different model sizes and training methods, whether pretrained or from scratch. Remarkably, without pretraining, ConvNova outperforms all Hyena-DNA models and DeepATT under the same epoch hyperparameter settings, even with a smaller parameter size.

Table 4: **Comparison of dilation and downsampling in ConvNova.** Performance (MCC-score or F1-score) is reported on selected NT benchmark tasks. The best values are in bold.

| Task | Downsampling (1.7M) | Dilation (1.7M) |
|---|---|---|
| H3 | 65.20 | **81.62** |
| H3K4me1 | 34.88 | **56.37** |
| H3K9ac | 48.81 | **68.98** |
| Promoter All | 93.69 | **96.99** |
| Splice Sites All | 45.65 | **96.45** |

### 4.3 ABLATION

### 4.4 DILATION OR DOWNSAMPLING

In image processing, the U-Net (Ronneberger et al., 2015) architecture has been proven highly effective through its use of downsampling and upsampling. Downsampling reduces image size while increasing the receptive field, enabling the capture of broader contextual information. Additionally, dilation, another method to increase the receptive field, expands the convolutional operation's receptive field by introducing gaps within the convolution kernel.

In DNA sequence modeling tasks, both the upsampling-downsampling structure and dilation convolution represent potentially viable approaches. For the dilation mechanism, we use the design shown in Figure 2. For the downsampling architecture, we implement a variant of ConvNova with U-Net style, maintaining the gating mechanism. We retain the same pretraining settings and parameter size (1.7M) and conduct comparative experiments on randomly selected tasks within the NT benchmark.

Due to the significant performance gap observed as shown in Table 4, we opt not to conduct multiple experiments using ten different random seeds. Instead, we only utilize a single random seed. It is evident that while the downsampling (U-Net style) architecture can approximate the performance achieved using dilation in the 'promoter all' task, it significantly lags behind in other tasks, even decreasing by 50 points in 'splice sites all'.

### 4.5 KEY DESIGNS

We conduct ablation experiments on the gate mechanism and the double-branch design in ConvNova. We implement an ordinary gate convolution model with the same parameter count (1.7M) named "Single Branch" in Table 5 to compare with the double-branch structure, and an additive aggregation model (1.7M) named "w/o Gate" in Table 5 to assess the role of the gate mechanism. We select homologous protein tasks from the NT benchmark and maintain the same method of selecting ten random seeds as described in §4.1.1 for comparison. For the details, refer to §A.3.1.

The results indicate that both the feature aggregation method and the gate

Table 5: **Ablation study results.** MCC-score ($\uparrow$) is reported for performance comparison across different models. 'w/o Gate' represents the ablation of the Gate mechanism, while 'Single Branch' is the ablation of the double-branch structure in ConvNova. The best values are in bold, and the second-best is underlined. ± indicates the error range across experiments

| Task | w/o Gate | Single Branch | ConvNova |
|---|---|---|---|
| H3 | 78.96 ±1.46 | 80.95 ±1.61 | **81.50** ±0.80 |
| H3K4me1 | **57.83** ±1.87 | 56.45 ±1.58 | 56.60 ±1.01 |
| H3K4me2 | 52.52 ±2.59 | 50.20 ±1.92 | **53.72** ±2.42 |
| H3K4me3 | 54.45 ±1.31 | 58.66 ±0.97 | **60.20** ±1.91 |
| H3K9ac | 63.54 ±2.20 | 66.87 ±0.69 | **68.10** ±1.91 |
| H3K14ac | 63.83 ±1.03 | 65.34 ±2.12 | **66.19** ±1.84 |
| H3K36me3 | 64.12 ±1.46 | 66.77 ±1.42 | **68.31** ±1.19 |
| H3K79me3 | 69.49 ±1.75 | 71.23 ±1.37 | **72.08** ±1.23 |
| H4 | 79.27 ±1.09 | 80.55 ±0.61 | **81.12** ±0.93 |
| H4ac | 62.02 ±2.28 | 63.07 ±1.10 | **64.75** ±1.90 |

mechanism in ConvNova are crucial design components, effectively supporting the overall model capability.

For detailed ablation on kernel size, dilation rate, please refer to §A.3.2.

## 5 DISCUSSION

We utilize dilation to control the receptive field in histone tasks from the NT-benchmark. H3K4me2, H3K4me3, and H3K14ac demonstrate significantly improved classification accuracy when the receptive field covers approximately 15% of the input length, compared to a global receptive field.

The enhancement effects of H3K14ac (Figure 6A) and H3K4me3 (Figure 6C) align with previous biological research (Ramakrishnan et al.,

Table 6: **ConvNova performance on Histone tasks in NT Benchmark.** Results are reported for models with 15% and 100% full sequence receptive fields. The best values are in bold.

| Task | 15% Receptive Field | 100% Receptive Field |
|---|---|---|
| H3K4me2 | **57.45** ±2.27 | 53.72 ±2.42 |
| H3K4me3 | **67.15** ±0.93 | 60.20 ±1.91 |
| H3K14ac | **70.71** ±2.32 | 66.19 ±1.84 |
| H3K9ac | 65.49 ±1.83 | **68.10** ±1.91 |

2016; Regadas et al., 2021), which has shown that these histone marks are highly enriched around transcription start sites (TSS), indicating their strong localized characteristics. As illustrated in Figure 6A, the pronounced enrichment of H3K14ac and H3K4me3 within this localized region supports the idea that a smaller receptive field is sufficient for effective classification in these tasks.

In contrast, H3K9ac (Figure 6D) exhibits a more uniform distribution across the gene body, which explains the suboptimal performance of a small receptive field in this case.

Interestingly, H3K4me2 (Figure 6B) also shows an enhancement effect with a small receptive field despite its enrichment being predominantly located in the middle of genes.

This observation raises two intriguing possibilities:

**Gene Position Context:** The enrichment of H3K4me2 in the middle of genes may indicate its role in transcriptional regulation during elongation. This localization could facilitate the binding of transcriptional machinery, enhancing transcriptional efficiency. Additionally, since our data were collected through ChIP-chip experiments on cells before and after oxidative stress, this enrichment might reflect interactions with genes associated with the histone methyltransferase Set1, suggesting a complex relationship that requires further exploration.

**Contextual Locality:** Alternatively, the improved performance of H3K4me2 in small receptive fields may not be tied to its position within genes but rather could stem from its involvement in other biological processes not captured during the experimental conditions. The sequences associated with H3K4me2 might exhibit strong local characteristics in contexts outside the immediate transcriptional environment, potentially involving regulatory mechanisms that are yet to be fully understood.

These possibilities highlight the need for further exploration to clarify the specific roles of H3K4me2 and its implications in various biological contexts.

## 6 CONCLUSION

In this study, we revisit the convolutional paradigm for DNA modeling. Through extensive analysis and experimentation, we show that convolutional neural networks maintain a competitive edge in DNA modeling tasks. This reevaluation prompts a reconsideration of CNNs, a relatively early method, for such functions within the community. We have introduced ConvNova, featuring three key design elements: dilated convolution, gated convolution, and a dual-branch structure. With these innovations, ConvNova achieves state-of-the-art performance on multiple benchmarks with small model sizes. Additionally, we have investigated the varying demands for receptive field sizes across different tasks. While some tasks exhibit expected behaviors, others may suggest the presence of undiscovered biological phenomena.

One **limitation** is that this work does not consider the multi-species pre-training dataset. Furthermore, we have not considered the specific genomic regions used in pretraining; it is possible that training exclusively in known functional regions could lead to improved performance. Furthermore, the current foundation model benchmarks are primarily limited to classification tasks, which results in a lack of diversity in the types of tasks evaluated. The **positive social impact** is that this work can accelerate DNA research. No **negative social impact** is perceived.

## 7 ACKNOWLEDGMENTS

This work was supported by the National Key R&D Program of China (NO.2022ZD0160101) and the National Natural Science Foundation of China (No. 62206244).

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

# A  APPENDIX

All the experiments are conducted with 4 RTX-4090 GPUs.

## A.1  PRETRAINING

We adhere to most of the pretraining hyperparameters and settings used for HyenaDNA, including the dataset selection. However, we implemented some small differences. We pretrained on sequences with a maximum length of 16K, while other pretraining sequence lengths were 1K or 2K, aligning with HyenaDNA's recommendation to use sequences 2-4 times longer than those required for downstream tasks. We set the global batch size to 512 and trained for 400 epochs with a learning rate of 1e-3. For MLM pretraining, selected nucleotides were "masked" by replacing them with "N" to predict the original nucleotide type. Table 7 and Table 8 illustrate the parameter settings for models of various sizes and the time needed to pretrain. The pretraining dataset is HG38[4].

Table 7: **Pretraining hyperparameters.** Values used during pretraining are reported.

| Hyperparameter | Value |
|---|---|
| Learning Rate | $1 \times 10^{-3}$ |
| Batch Size | 256 |
| Weight Decay | 0.1 |
| Dropout | 0.0 |
| Optimizer | AdamW |
| Optimizer Momentum | $\beta_1 = 0.9, \beta_2 = 0.999$ |
| Learning Rate Scheduler | Cosine Decay |
| Training Epochs | 200 |

Table 8: **Pretraining time.** The time required for pretraining different model sizes with varying sequence lengths on 4 RTX-4090 GPUs is reported.

| Parameters | 386K | 1.7M | 7.4M | 27.4M |
|---|---|---|---|---|
| Sequence Length | 1K | 1K | 16K | 2K |
| Time | 50 mins | 80 mins | 14 hours | 4 hours |

## A.2  DOWNSTREAM TASKS

All hyperparameters for the ConvNova model on downstream tasks can be found in Table 9.

Table 9: **Hyperparameters for ConvNova model on all downstream tasks.** Specific settings for each task include sequence length, dilation rate, hidden dimension, number of GCBs, and model size.

| Tasks | Sequence Length | Dilation Rate | Hidden Dim | Num GCB | Size |
|---|---|---|---|---|---|
| NT | 1K | 4/1 | 128 | 5 | 1.7M |
| Deepstarr | 1K | 4 | 128 | 5 | 1.7M |
| Genomic Benchmark | 1K | 4 | 64 | 5 | 386K |
| Chromatin Profile Prediction | 2K | 4 | 512/256 | 5 | 27.4M/6.8M |
| Gene Finding | 16K | 4 | 196 | 10 | 7.4M |

### A.2.1  NUCLEOTIDE TRANSFORMER BENCHMARK

**Objective:** Sequence classification.

---

[4]https://www.ncbi.nlm.nih.gov/assembly/GCF_000001405.26

**Models and Setup:** We follow the HyenaDNA setup (Nguyen et al., 2024), splitting the dataset into 90/10 training and test sets and fine-tuning DNABERT-2, HyenaDNA, NT-v2, and Caduceus-Ph for 20 epochs using pre-trained weights from Hugging Face.

**Experiment Configuration:** All models are tested with 32-bit floating point precision, and ConvNova uses consistent hyperparameters across tasks as detailed in Table 9, with ten random seeds to report mean values, shown in Tables 16 and 17. Optimizer we use AdamW and share the same setting as Table 7. Learning rate we use 1e-3 for ConvNova and official learning rate for the baselines.

### A.2.2    CHROMATIN PROFILE PREDICTION

**Objective:** Perform 919 binary classification tasks on DNA fragments using 4.4 million training data points. Even small improvements (<1%) are significant for this challenging task.

**Models and Setup:** The baseline models include HyenaDNA (0.4M, 3.3M, 6.5M), NT-v2 (50M, 100M, 250M, 500M), DNABERT-2, and DeepATT (previous state-of-the-art CNN for this task), trained for 10 epochs. All models are tested using 32-bit floating point precision.

**Experiment Configuration:** The train/test split follows the DeepSEA paper methodology. The optimizer used is AdamW with settings from Table 7, and a learning rate of 1e-3 for ConvNova, with official learning rates used for the baselines.

We provide more detailed comparisons of different models in Table 10. Besides the pretrained ConvNova model, we also train ConvNova from scratch. Remarkably, even without pretraining, the model outperformed all versions of HyenaDNA at similar parameter sizes, further demonstrating ConvNova's superior ability to encode long-sequence information. Hyperparameters can be found in Table 9.

Table 10: **Chromatin Profile Prediction detailed comparison.** AUC Score (↑) is reported for ConvNova, HyenaDNA, NT v2, DNABERT-2, DeepATT performance on transcription factors (TF), DNase I hypersensitive sites (DHS), histone modifications (HM), and average score. The best values per task are bolded, and the second best are underlined.

| Tasks | Pretrain | | | | | | | | | | From Scratch | | | |
|---|---|---|---|---|---|---|---|---|---|---|---|---|---|---|
| | ConvNova | | Hyena-DNA | | | NTv2 | | | | DNABERT-2 | ConvNova | | | DeepATT |
| | 6.8M | 27.4M | 0.4M | 3.3M | 6.5M | 50M | 100M | 250M | 500M | 117M | 1.7M | 6.8M | 27.4M | 7.8M |
| TF | 96.77 | **96.99** | 94.42 | 94.72 | 95.91 | 96.17 | 96.23 | 96.64 | 96.76 | 96.16 | 96.62 | 96.63 | 96.77 | 96.35 |
| HM | 86.54 | 86.70 | 84.64 | 84.96 | 85.46 | 86.23 | 86.20 | **86.81** | 86.76 | 86.18 | 85.74 | 85.74 | 85.76 | 86.16 |
| DHS | 93.42 | **93.71** | 90.58 | 90.90 | 92.33 | 92.50 | 92.65 | 93.13 | 93.39 | 92.50 | 92.93 | 92.94 | 93.03 | 92.52 |
| Avg. | 92.25 | **92.47** | 89.89 | 90.19 | 91.23 | 91.63 | 91.69 | 92.19 | 92.30 | 91.61 | 91.76 | 91.77 | 91.85 | 91.68 |

### A.2.3    GENEFINDING

**Objective:** Classify each nucleotide based on its gene structure context, predicting exon-intron boundaries and capturing long-range dependencies for accurate annotation.

**Models and Setup:** The baseline models include NT-H, DNABERT2, HyenaDNA, and GENA_LM. ConvNova is tuned for 10 epochs, while the results of the other baselines are taken from BEND.

**Experiment Configuration:** The optimizer used is AdamW with settings from Table 7, and a learning rate of 1e-3 for ConvNova.

**Dilation and Layer impact on Gene Finding**    We find that the gene-finding task requires long-range dependency. Both decreasing dilation rate and the number of layers can result in performance decline in Figure 4. Models are trained from scratch in all experiments. For Dilation impact, the model dimension is 256 and contains 5 GCBs. For GCB counts impact, the model dimension is set to 256, and the dilation rate is set to 4. For additional hyperparameter settings in this task, readers can refer to Table 9.

### A.2.4    GENOMIC BENCHMARK

**Objective:** Sequence classification task.

Figure 4: **The impact of dilation rate and GCB counts on the MCC in ConvNova model.**

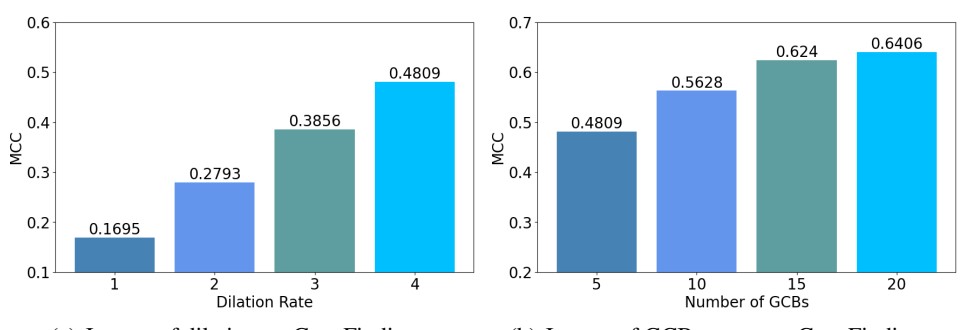

(a) Impact of dilation on GeneFinding.    (b) Impact of GCB counts on GeneFinding.

**Models and Setup:** All models are fine-tuned for ten epochs. ConvNova is used as the backbone with a width of 64, 5 GCBs, and a parameter size of 386K, pretrained on sequences of length 1000. The baselines used in this task are CaduceusPh, HyenaDNA, and the CNN model provided in the benchmark. We follow the train-valid split (90/10) as provided by HyenaDNA (Nguyen et al., 2024).

**Experiment Configuration:** The models are trained with 16-precision float points. We use five random seeds and report the mean result across all models. Hyperparameters for this task can be found in Table 9.The optimizer used is AdamW with settings from Table 7, and a learning rate of 1e-3 for ConvNova, with official learning rates used for the baselines.

### A.3 Ablations experiments

#### A.3.1 Gating Mechanism and Dual Branch

In this section, we provide more details about the ablation experiments. There are two variants of the ConvNova model: "Single Gate" and "Dual Branch", and one variant without any gating mechanism.

**Single Gate** Our "Single Gate" design follows a sequential process as shown in Eq.2. First, a single `Conv1D` layer processes the input feature $\mathbf{A} \in \mathbb{R}^{l \times d}$ with weight matrix $\mathbf{W}_d \in \mathbb{R}^{k \times d \times d}$ and bias term $\mathbf{b}_d \in \mathbb{R}^d$ to produce $\mathbf{z} \in \mathbb{R}^{l \times d}$. This output is then processed through two parallel activation paths - `GELU` activation and `sigmoid` activation ($\sigma$). The final output is obtained through residual addition of element-wise multiplication ($\odot$) of activated $\mathbf{h}$ and gated $\mathbf{g}$. To maintain parameter count parity with ConvNova, we increase the dimension of the `Conv1D` layer.

$$
\begin{aligned}
\mathbf{z} &= \mathbf{W}_d * \mathbf{A} + \mathbf{b}_d \\
\mathbf{h} &= \mathrm{GELU}(\mathbf{z}) \\
\mathbf{g} &= \sigma(\mathbf{z}) \\
\mathbf{A} &= \mathbf{A} + \mathbf{h} \odot \mathbf{g}
\end{aligned}
\tag{2}
$$

**Dual Gate** Our "Dual Branch" design processes input features through two parallel convolutional paths with gating mechanisms, as shown in Eq.3. The inputs $\mathbf{A}, \mathbf{B} \in \mathbb{R}^{l \times d}$ are processed through two separate `Conv1D` layers with weights $\mathbf{W}_{1,d/\sqrt{2}}, \mathbf{W}_{2,d/\sqrt{2}} \in \mathbb{R}^{k \times d \times d/\sqrt{2}}$ and biases $\mathbf{b}_{1,d/\sqrt{2}}, \mathbf{b}_{2,d/\sqrt{2}} \in \mathbb{R}^{d/\sqrt{2}}$ to produce intermediate features $\mathbf{z}_1, \mathbf{z}_2 \in \mathbb{R}^{l \times d/\sqrt{2}}$. The first path applies `GELU` activation to $\mathbf{z}_1$, while the second applies `sigmoid` activation to $\mathbf{z}_2$. The final output $\mathbf{A}$ is obtained through residual addition of element-wise multiplication ($\odot$) of activated $\mathbf{h}$ and gated $\mathbf{g}$, and $\mathbf{B}$ is obtained through residual addition of gated $\mathbf{g}$.

$$
\begin{aligned}
\mathbf{z}_1 &= \mathbf{W}_{1,d/\sqrt{2}} * \mathbf{A} + \mathbf{b}_{1,d/\sqrt{2}} \\
\mathbf{z}_2 &= \mathbf{W}_{2,d/\sqrt{2}} * \mathbf{B} + \mathbf{b}_{2,d/\sqrt{2}} \\
\mathbf{h} &= \mathrm{GELU}(\mathbf{z}_1) \\
\mathbf{g} &= \sigma(\mathbf{z}_2) \\
\mathbf{A} &= \mathbf{A} + \mathbf{h} \odot \mathbf{g} \\
\mathbf{B} &= \mathbf{B} + \mathbf{g}
\end{aligned} \tag{3}
$$

**Addition w/o Gate** This variant uses simple addition operations for feature aggregation without any gating mechanism. As shown in Eq.4, the input features $\mathbf{A}, \mathbf{B} \in \mathbb{R}^{l \times d}$ are processed through convolutional layers and GELU activation to produce intermediate features $\mathbf{h}, \mathbf{g} \in \mathbb{R}^{l \times d/\sqrt{2}}$. These are then combined through addition to produce the final outputs $\mathbf{A}$ and $\mathbf{B}$.

$$
\begin{aligned}
\mathbf{h} &= \mathrm{GELU}(\mathbf{W}_{1,d/\sqrt{2}} * \mathbf{A} + \mathbf{b}_{1,d/\sqrt{2}}) \\
\mathbf{g} &= \mathrm{GELU}(\mathbf{W}_{2,d/\sqrt{2}} * \mathbf{B} + \mathbf{b}_{2,d/\sqrt{2}}) \\
\mathbf{A} &= \mathbf{A} + \mathbf{h} + \mathbf{g} \\
\mathbf{B} &= \mathbf{B} + \mathbf{g}
\end{aligned} \tag{4}
$$

### A.3.2 Dilation Mechanism, Kernel Size and local dependency analysis

In this section, we present the results of the ablation experiments on the dilation rate and kernel size. We perform experiments on the NT benchmark, testing kernel sizes of 3, 5, 7, 9, 11 and dilation rates of 1, 2, 3, 4. We report the average results across 18 tasks from the entire benchmark.

As shown in Table 11, the performance generally improves with increasing dilation rate and kernel size. However, to balance compatibility with long-range tasks and maintain relatively small model parameters, we ultimately select a kernel size of 9 and a dilation rate of 4.

Table 11: Performance of different kernel sizes and dilation rates across 18 tasks in the NT benchmark. The values represent the average performance for all tasks.

| Kernel Size \Dilation Rate | 1 | 2 | 3 | 4 |
|---|---|---|---|---|
| 3 | 63.710 | 67.440 | 69.630 | 71.705 |
| 5 | 63.939 | 70.324 | 73.049 | 73.688 |
| 7 | 68.353 | 71.141 | 74.049 | 74.598 |
| 9 | 68.959 | 72.257 | 74.241 | 74.530 |
| 11 | 69.408 | 73.437 | 74.835 | 74.549 |

Furthermore, we observe that even for short-range tasks (all having a sequence length of 500 bp), the intensity of local dependencies varies significantly. For instance, the tasks H3K4me2, H3K4me3, and H3K14ac, as discussed in §5, exhibit strong local dependencies, with better performance observed when the dilation rate is small. On the other hand, tasks like splice site prediction show a much stronger global dependency, which benefits from a larger dilation rate. See Table 12 for more details.

### A.4 Supervised Methods against Foundation Models

We conduct a comparison between previous supervised models and DNA foundation models on both the NT benchmark and genomic benchmark. For our comparison, we select the popular Basenji and the recently established state-of-the-art model, LegNet, which excels in short DNA regulatory regions. Originally, these supervised models are designed for specific tasks. We apply them to the benchmarks used for foundation models. As observed in Table 13 and Table 14, although they show decent performance on specific tasks, they do not achieve the same level of consistency across various datasets as foundation models do. This inconsistency may arise from the enhanced modeling capabilities afforded by pretraining in foundation models.

Table 12: Performance of different dilation rates and kernel sizes on the NT benchmark. For example, `k3_d1` represents a kernel size of 3 and a dilation rate of 1, with similar notation for other configurations.

| Task | | | | Histone Modifications | | | | |
| --- | --- | --- | --- | --- | --- | --- | --- | --- |
| | H3 | H3K4me1 | H3K4me2 | H3K4me3 | H3K9ac | H3K14ac | H3K36me3 | H3K79me3 |
| k3_d1 | 70.14 | 43.14 | 41.26 | 49.12 | 54.00 | 54.80 | 54.48 | 62.38 |
| k3_d2 | 74.66 | 50.54 | 47.52 | 56.50 | 58.92 | 60.16 | 63.18 | 67.50 |
| k3_d3 | 79.24 | 52.44 | 49.50 | 55.18 | 61.60 | 62.72 | 63.66 | 68.50 |
| k3_d4 | 78.72 | 51.94 | 47.54 | 54.12 | 62.36 | 62.30 | 63.74 | 68.48 |
| k5_d1 | 70.76 | 50.26 | 48.28 | 56.86 | 58.02 | 61.62 | 60.44 | 65.38 |
| k5_d2 | 76.80 | 55.88 | 53.50 | 62.30 | 63.58 | 66.24 | 65.04 | 70.30 |
| k5_d3 | 80.88 | 55.58 | 52.16 | 58.86 | 65.62 | 65.22 | 66.02 | 70.84 |
| k5_d4 | 80.48 | 55.12 | 50.46 | 56.56 | 65.60 | 63.58 | 65.58 | 69.70 |
| k7_d1 | 71.76 | 53.12 | 50.72 | 59.78 | 61.06 | 65.16 | 62.80 | 67.32 |
| k7_d2 | 77.72 | 56.62 | 53.66 | 62.54 | 65.96 | 66.92 | 66.06 | 70.78 |
| k7_d3 | 81.20 | 56.42 | 53.38 | 58.04 | 66.56 | 65.30 | 66.54 | 70.96 |
| k7_d4 | 81.62 | 56.46 | 51.02 | 57.60 | 66.38 | 64.24 | 65.96 | 69.10 |
| k9_d1 | 72.52 | 54.70 | 52.90 | 60.50 | 62.26 | 66.02 | 63.94 | 69.14 |
| k9_d2 | 78.02 | 56.76 | 55.38 | 62.04 | 66.80 | 67.46 | 67.62 | 71.52 |
| k9_d3 | 81.58 | 56.16 | 52.20 | 58.64 | 66.70 | 65.00 | 66.28 | 71.84 |
| k9_d4 | 81.18 | 55.36 | 48.54 | 56.60 | 65.84 | 64.40 | 66.08 | 70.02 |
| k11_d1 | 72.60 | 55.90 | 53.78 | 62.80 | 63.02 | 66.40 | 66.14 | 69.58 |
| k11_d2 | 79.62 | 56.74 | 55.20 | 63.84 | 67.08 | 67.24 | 68.20 | 71.54 |
| k11_d3 | 81.66 | 56.12 | 53.98 | 58.66 | 66.88 | 64.62 | 66.84 | 71.22 |
| k11_d4 | 80.66 | 55.16 | 47.84 | 55.76 | 66.82 | 63.20 | 66.28 | 70.18 |

| Task | promoter | | | splice_sites | | | enhancer | |
| --- | --- | --- | --- | --- | --- | --- | --- | --- |
| | all | non_tata | tata | all | acceptor | donor | | types |
| k3_d1 | 94.68 | 94.82 | 94.54 | 47.08 | 79.72 | 78.16 | 54.04 | 48.96 |
| k3_d2 | 96.06 | 96.04 | 95.74 | 49.26 | 80.62 | 79.84 | 52.36 | 49.76 |
| k3_d3 | 96.50 | 96.44 | 96.20 | 60.36 | 84.10 | 83.10 | 53.84 | 50.66 |
| k3_d4 | 96.62 | 96.64 | 96.38 | 91.42 | 87.54 | 87.12 | 56.46 | 48.96 |
| k5_d1 | 95.10 | 95.30 | 94.88 | 48.20 | 79.84 | 78.20 | 52.44 | 51.30 |
| k5_d2 | 96.36 | 96.48 | 96.32 | 53.84 | 82.22 | 81.28 | 52.96 | 50.50 |
| k5_d3 | 96.68 | 96.80 | 96.54 | 88.70 | 86.96 | 85.74 | 55.68 | 49.62 |
| k5_d4 | 96.76 | 96.74 | 96.18 | 96.38 | 96.08 | 95.62 | 56.14 | 51.84 |
| k7_d1 | 95.36 | 95.46 | 95.46 | 48.50 | 79.72 | 78.92 | 55.70 | 51.48 |
| k7_d2 | 96.54 | 96.48 | 96.32 | 59.54 | 83.76 | 83.12 | 53.26 | 49.00 |
| k7_d3 | 96.76 | 96.70 | 96.28 | 95.38 | 88.86 | 88.82 | 57.42 | 49.88 |
| k7_d4 | 96.76 | 96.68 | 96.48 | 96.24 | 97.02 | 96.40 | 58.02 | 50.02 |
| k9_d1 | 95.56 | 95.66 | 95.28 | 49.16 | 80.36 | 79.40 | 53.58 | 49.60 |
| k9_d2 | 96.54 | 96.54 | 96.34 | 70.96 | 84.78 | 84.16 | 54.06 | 48.12 |
| k9_d3 | 96.70 | 96.66 | 96.40 | 95.94 | 91.98 | 90.98 | 56.82 | 48.94 |
| k9_d4 | 96.74 | 96.82 | 96.92 | 96.54 | 97.08 | 96.48 | 58.60 | 50.00 |
| k11_d1 | 95.68 | 95.84 | 95.06 | 49.08 | 80.78 | 79.04 | 54.56 | 49.18 |
| k11_d2 | 96.40 | 96.46 | 96.40 | 84.60 | 85.62 | 84.84 | 55.92 | 48.20 |
| k11_d3 | 96.64 | 96.58 | 96.22 | 95.84 | 96.86 | 95.62 | 57.16 | 48.58 |
| k11_d4 | 96.82 | 96.66 | 96.60 | 96.76 | 97.16 | 86.46 | 61.68 | 50.80 |

Table 13: **Supervised Methods against Foundation Models on NT Benchmark.** MCC/F1-score is reported for pretrained NTv2, HyenaDNA, DNABERT-2, Caduceus-Ph, ConvNova, Basenji, LegNet ConvNova*(* means trained from scratch)

. The best values per task are bold, and the second-best are underlined. ± indicates the error range across five random seeds.

| | NTv2 (500M) | HyenaDNA (1.6M) | DNABERT-2 (117M) | Caduceus-Ph (1.9M) | ConvNova (1.7M) | Basenji (7.4M) | LegNet (2.1M) | ConvNova* (1.7M) |
|---|---|---|---|---|---|---|---|---|
| *Histone* | | | | | | | | |
| H3 | 78.17 ±2.54 | 78.14 ±1.70 | 79.31 ±0.68 | 80.48 ±1.04 | **81.50** ±0.80 | 78.05 ±1.83 | 76.24 ±0.83 | 81.18 ±1.68 |
| H3K4me1 | 51.64 ±1.12 | 44.52 ±2.59 | 48.34 ±4.63 | 52.83 ±0.96 | **56.60** ±1.01 | 42.70 ±3.16 | 47.47 ±1.36 | 55.36 ±2.26 |
| H3K4me2 | 37.24 ±2.25 | 42.68 ±2.66 | 43.02 ±2.92 | 49.88 ±2.65 | **57.45** ±2.27 | 34.73 ±1.66 | 45.75 ±0.56 | 48.54 ±3.24 |
| H3K4me3 | 50.30 ±1.77 | 50.41 ±3.15 | 45.43 ±3.33 | 56.72 ±2.58 | **67.15** ±0.93 | 38.85 ±1.49 | 52.67 ±2.54 | 56.60 ±2.70 |
| H3K9ac | 61.05 ±1.40 | 58.50 ±1.75 | 60.04 ±1.27 | 63.27 ±2.29 | **68.10** ±1.91 | 56.34 ±2.86 | 58.21 ±0.92 | 65.84 ±1.14 |
| H3K14ac | 57.22 ±2.19 | 56.71 ±2.40 | 54.49 ±4.99 | 60.84 ±2.94 | **70.71** ±2.32 | 49.88 ±1.60 | 56.64 ±2.12 | 64.40 ±0.90 |
| H3K36me3 | 60.50 ±1.75 | 59.92 ±1.06 | 57.58 ±2.38 | 61.12 ±1.44 | **68.31** ±1.19 | 52.73 ±1.84 | 57.54 ±1.03 | 66.08 ±1.88 |
| H3K79me3 | 65.78 ±2.34 | 66.25 ±3.65 | 64.38 ±0.48 | 67.17 ±2.03 | **72.08** ±1.23 | 62.43 ±1.29 | 63.78 ±1.72 | 70.02 ±0.72 |
| H4 | 79.87 ±1.34 | 78.15 ±1.58 | 78.18 ±0.98 | 80.10 ±1.00 | **81.12** ±0.93 | 77.60 ±2.02 | 78.73 ±1.93 | 80.78 ±0.82 |
| H4ac | 55.22 ±2.20 | 54.15 ±2.96 | 51.80 ±0.10 | 59.26 ±3.67 | **66.10** ±1.20 | 46.05 ±1.77 | 55.29 ±2.60 | 63.56 ±1.14 |
| *Regulatory* | | | | | | | | |
| Enhancer | 54.51 ±1.94 | 53.13 ±4.52 | 52.50 ±1.44 | 55.20 ±2.56 | **57.60** ±2.52 | 57.54 ±2.75 | 55.25 ±3.23 | 58.60 ±3.80 |
| Enhancer Types | 43.36 ±1.75 | 48.16 ±2.48 | 44.32 ±1.18 | 47.17 ±2.85 | 49.75 ±2.82 | **52.67** ±2.94 | 44.72 ±2.04 | 50.00 ±1.80 |
| Promoter All | **96.82** ±0.47 | 95.57 ±0.18 | 96.23 ±0.17 | 96.65 ±0.16 | **96.82** ±0.22 | 95.79 ±0.24 | 96.52 ±0.33 | 96.74 ±0.06 |
| Promoter non-TATA | **97.45** ±0.69 | 95.86 ±0.37 | 97.17 ±0.17 | 96.31 ±0.50 | 96.76 ±0.21 | 95.99 ±0.18 | 96.45 ±0.27 | 96.82 ±0.28 |
| Promoter TATA | 96.53 ±0.81 | 95.88 ±0.53 | **96.99** ±0.49 | 96.21 ±0.81 | 96.34 ±0.38 | 95.89 ±0.78 | 96.25 ±0.45 | 96.92 ±0.32 |
| *Splice sites* | | | | | | | | |
| Splice Acceptor | **97.99** ±0.66 | 96.98 ±0.49 | 97.49 ±0.36 | 94.21 ±0.37 | 96.23 ±0.41 | 97.65 ±0.34 | 85.80 ±0.58 | 97.08 ±0.32 |
| Splice Donor | **98.50** ±0.43 | 95.27 ±1.07 | 94.33 ±0.27 | 94.69 ±0.67 | 96.62 ±0.61 | 97.82 ±0.66 | 84.72 ±0.95 | 96.48 ±0.32 |
| Splice All | **98.15** ±1.01 | 94.05 ±1.08 | 93.75 ±1.25 | 92.87 ±1.73 | 96.33 ±0.31 | 98.07 ±0.45 | 54.71 ±2.38 | 96.54 ±0.36 |

Table 14: **Supervised Methods against Foundation Models on Genomic Benchmark.** Top-1 accuracy (↑) is reported for pretrained HyenaDNA, Caduceus-Ph, ConvNova, Basenji, LegNet, and the original CNN baseline (trained from scratch). The best values are in bold, and the second-best is underlined. ± indicates the error range across five random seeds.

| Task | CNN (264K) | HyenaDNA (436K) | Caduceus-Ph (470K) | ConvNova (386K) | Basenji (7.4N) | LegNet (2.1M) |
|---|---|---|---|---|---|---|
| *Enhancers* | | | | | | |
| Mouse Enhancers | 0.730 ±0.032 | 0.779 ±0.013 | 0.754 ±0.074 | **0.784** ±0.009 | 0.659 ±0.155 | 0.504 ±0.000 |
| Human Enhancers Cohn | 0.702 ±0.021 | 0.718 ±0.008 | **0.747** ±0.004 | 0.743 ±0.005 | 0.712 ±0.030 | 0.739 ±0.004 |
| Human Enhancer Ensembl | 0.744 ±0.122 | 0.832 ±0.006 | 0.893 ±0.008 | 0.900 ±0.004 | **0.905** ±0.007 | 0.879 ±0.002 |
| *Species Classification* | | | | | | |
| Coding vs. Intergenomic | 0.892 ±0.008 | 0.904 ±0.008 | 0.915 ±0.003 | **0.943** ±0.001 | 0.905 ±0.004 | 0.939 ±0.002 |
| Human vs. Worm | 0.942 ±0.002 | 0.961 ±0.002 | **0.973** ±0.001 | 0.967 ±0.002 | 0.957 ±0.003 | 0.965 ±0.001 |
| *Regulatory Elements* | | | | | | |
| Human Regulatory | 0.872 ±0.005 | 0.862 ±0.004 | 0.872 ±0.011 | **0.873** ±0.002 | 0.764 ±0.005 | 0.764 ±0.006 |
| Human Non-TATA Promoters | 0.861 ±0.009 | 0.887 ±0.005 | 0.946 ±0.007 | **0.951** ±0.003 | 0.919 ±0.006 | 0.942 ±0.007 |
| Human OCR Ensembl | 0.698 ±0.013 | 0.744 ±0.019 | **0.828** ±0.006 | 0.793 ±0.004 | 0.766 ±0.009 | 0.802 ±0.004 |

## A.5 NEIGHBORHOOD IMPORTANCE IN DNA MODELING

To support our hypothesis that the inductive bias of CNNs—emphasizing neighboring nucleotides—is beneficial for DNA modeling, we make modifications to the Rotary Position Embedding (RoPE) mechanism. Specifically, we adjust the original $\theta$ values, enabling each attention head to have distinct $\theta$ values.

Additionally, we modified the initialization of the bias term in the linear layers of the $K$ and $Q$ projections. The bias was initialized to $[0, 0, 0, 0, 0, \ldots, 1]$, while all other parameters retained the initialization strategy of NTv2 (standard deviation $\sigma = 0.02$, mean $\mu = 0$). This adjustment ensures that the initialization process places greater emphasis on neighboring nucleotides. Refer to Figure A.5 for an illustration of the improved attention map, which places greater emphasis on neighboring tokens.

To validate our hypothesis, we train NTv2 from scratch on three tasks where ConvNova with small dilation values performs better (H3K4me2, H3K4me3, and H3K14ac). The results, presented in Table

15, show that the adjusted NTv2 consistently outperform its original version. However, despite these improvements, our naive modifications to the transformer architecture does not surpass ConvNova trained from scratch(See Table 13). To enhance transformer and state space model (SSM) designs for DNA modeling further dedicated efforts is needed.

Our primary goal here is to validate the hypothesis and provide an explanation for why CNNs might outperform transformers in this context.

Table 15: Performance comparison of NTv2 and NTv2*, where NTv2* represents our modified version. Results demonstrate that NTv2* achieves significant improvements.

| Task | NTv2* | NTv2 |
|------|-------|------|
| H3K14ac | 46.65 | 34.42 |
| H3K4me2 | 33.10 | 25.79 |
| H3K4me3 | 34.62 | 21.40 |

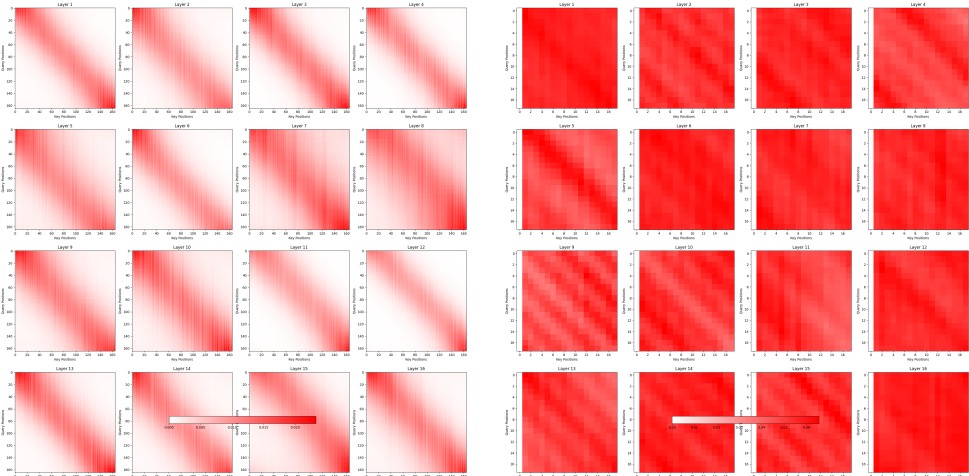

Figure 5: Visualization of the attention maps for NTv2* (Left) and NTv2 (Right). The asterisk (*) denotes the modified version. The modified initialization places more emphasis on neighboring tokens in the attention map.

Table 16: **Results for all models on NT benchmark across genomic regions.** Ten random seeds were used for each model. The best values per task are bolded.

| Seed | 2222 | 42 | 43 | 44 | 45 | 46 | 47 | 48 | 49 | 50 |
|---|---|---|---|---|---|---|---|---|---|---|
| *Promoter All* | | | | | | | | | | |
| ConvNova (4 dilation) | 96.99 | 96.84 | 96.81 | 96.82 | 96.75 | 96.82 | 96.64 | 96.60 | 96.96 | 96.96 |
| Caduceus-Ph | 96.66 | 96.61 | 96.64 | 96.61 | 96.51 | 96.79 | 96.77 | 96.49 | 96.73 | 96.68 |
| HyenaDNA | 95.55 | 95.62 | 95.44 | 95.58 | 95.62 | 95.58 | 95.56 | 95.65 | 95.75 | 95.39 |
| DNABERT-2 | 96.38 | 96.27 | 96.15 | 96.21 | 96.05 | 96.19 | 96.31 | 96.22 | 96.29 | 96.11 |
| NTv2 | 96.47 | 97.19 | 96.95 | 96.35 | 97.12 | 96.63 | 97.21 | 96.85 | 96.40 | 97.00 |
| *Promoter Non-TATA* | | | | | | | | | | |
| ConvNova (4 dilation) | 96.55 | 96.81 | 96.73 | 96.83 | 96.88 | 96.61 | 96.77 | 96.83 | 96.87 | 96.72 |
| Caduceus-Ph | 96.14 | 95.81 | 96.14 | 96.52 | 96.79 | 95.98 | 96.47 | 96.36 | 96.39 | 96.50 |
| HyenaDNA | 95.81 | 96.23 | 95.66 | 96.00 | 95.96 | 95.60 | 95.63 | 96.04 | 95.87 | 95.83 |
| DNABERT-2 | 97.18 | 97.02 | 97.31 | 97.22 | 97.04 | 97.29 | 97.15 | 97.03 | 97.26 | 97.12 |
| NTv2 | 96.89 | 97.44 | 98.01 | 96.81 | 97.29 | 98.14 | 97.52 | 96.78 | 97.93 | 97.66 |
| *Promoter TATA* | | | | | | | | | | |
| ConvNova (4 dilation) | 96.57 | 96.08 | 95.92 | 96.57 | 96.07 | 96.40 | 96.57 | 96.09 | 96.41 | 96.72 |
| Caduceus-Ph | 95.40 | 96.72 | 96.40 | 95.73 | 96.23 | 96.74 | 96.24 | 95.71 | 96.56 | 96.41 |
| HyenaDNA | 95.92 | 96.06 | 96.23 | 95.89 | 96.41 | 95.55 | 95.58 | 96.16 | 95.58 | 95.42 |
| DNABERT-2 | 97.38 | 97.12 | 96.52 | 97.41 | 97.18 | 96.53 | 97.40 | 97.15 | 96.54 | 96.87 |
| NTv2 | 95.75 | 96.89 | 97.13 | 95.92 | 96.48 | 96.12 | 97.05 | 96.73 | 95.88 | 97.34 |
| *Splice Sites All* | | | | | | | | | | |
| ConvNova (4 dilation) | 96.46 | 96.18 | 96.6 | 96.41 | 96.28 | 96.50 | 95.82 | 96.30 | 95.93 | 95.85 |
| Caduceus-Ph | 96.29 | 96.33 | 96.21 | 96.36 | 96.58 | 96.04 | 96.19 | 95.84 | 96.03 | 96.27 |
| HyenaDNA | 96.97 | 96.99 | 96.93 | 96.74 | 96.96 | 97.22 | 97.05 | 96.89 | 97.47 | 96.61 |
| DNABERT-2 | 94.32 | 94.18 | 93.71 | 94.29 | 94.21 | 93.72 | 94.34 | 94.23 | 93.74 | 94.03 |
| NTv2 | 97.14 | 98.47 | 98.97 | 97.88 | 98.63 | 98.09 | 97.57 | 98.42 | 98.28 | 98.05 |
| *Splice Sites Acceptor* | | | | | | | | | | |
| ConvNova (4 dilation) | 96.65 | 96.01 | 96.87 | 96.70 | 96.86 | 96.59 | 96.37 | 97.06 | 96.50 | 96.59 |
| Caduceus-Ph | 97.36 | 96.41 | 96.74 | 96.61 | 96.24 | 96.83 | 96.74 | 96.87 | 96.50 | 96.56 |
| HyenaDNA | 94.20 | 95.37 | 95.21 | 94.97 | 96.06 | 95.94 | 95.2 | 94.48 | 95.73 | 95.56 |
| DNABERT-2 | 94.57 | 94.21 | 94.23 | 94.55 | 94.19 | 94.24 | 94.59 | 94.18 | 94.25 | 94.28 |
| NTv2 | 98.57 | 97.89 | 98.14 | 97.45 | 98.64 | 97.72 | 98.09 | 97.33 | 98.38 | 97.69 |
| *Splice Sites Donor* | | | | | | | | | | |
| ConvNova (4 dilation) | 96.57 | 96.34 | 96.52 | 96.51 | 96.44 | 96.02 | 96.04 | 96.38 | 96.12 | 96.33 |
| Caduceus-Ph | 92.47 | 93.21 | 94.74 | 92.61 | 92.78 | 93.56 | 91.14 | 94.36 | 92.57 | 91.22 |
| HyenaDNA | 94.38 | 94.59 | 95.04 | 93.09 | 94.63 | 93.38 | 94.52 | 93.67 | 92.97 | 94.22 |
| DNABERT-2 | 92.51 | 94.76 | 93.72 | 92.48 | 94.81 | 93.74 | 92.53 | 94.79 | 93.71 | **94.52** |
| NTv2 | 98.93 | 98.67 | 98.43 | 98.12 | 98.54 | 98.79 | 98.26 | 98.68 | 98.37 | 98.21 |
| *Enhancer* | | | | | | | | | | |
| ConvNova (4 dilation) | 59.00 | 55.08 | 57.09 | 58.54 | 55.33 | 58.26 | 58.77 | 58.42 | 58.10 | 57.39 |
| Caduceus-Ph | 53.72 | 53.64 | 54.04 | 53.50 | 57.76 | 56.75 | 55.94 | 55.35 | 54.66 | 56.63 |
| HyenaDNA | 55.34 | 53.88 | 54.43 | 51.44 | 51.56 | 54.28 | 55.53 | 48.61 | 53.18 | 53.05 |
| DNABERT-2 | 52.11 | 51.06 | 52.15 | 53.67 | 53.23 | 53.24 | 51.56 | 52.08 | 52.88 | 53.06 |
| NTv2 | 53.25 | 55.80 | 54.10 | 52.99 | 56.45 | 54.75 | 53.40 | 55.50 | 54.99 | 53.85 |
| *Enhancer Types* | | | | | | | | | | |
| ConvNova (4 dilation) | 49.08 | 48.32 | 49.24 | 52.56 | 48.72 | 48.90 | 51.24 | 49.45 | 49.96 | 49.98 |
| Caduceus-Ph | 47.78 | 44.58 | 45.93 | 46.01 | 48.04 | 50.02 | 47.98 | 47.09 | 46.93 | 47.35 |
| HyenaDNA | 47.71 | 47.77 | 48.06 | 48.49 | 50.34 | 49.67 | 49.45 | 48.43 | 45.68 | 46.09 |
| DNABERT-2 | 43.68 | 44.17 | 43.29 | 45.50 | 43.23 | 45.29 | 44.22 | 45.18 | 45.40 | 43.28 |
| NTv2 | 42.15 | 44.90 | 43.20 | 41.85 | 44.25 | 43.80 | 42.50 | 45.10 | 43.05 | 42.75 |

Table 17: **(Cont.) Results for all models on NT benchmark across genomic regions.**

| Task/Model | 2222 | 42 | 43 | 44 | 45 | 46 | 47 | 48 | 49 | 50 |
|---|---|---|---|---|---|---|---|---|---|---|
| **H3** | | | | | | | | | | |
| ConvNova (4 dilation) | 81.62 | 81.78 | 81.04 | 82.24 | 82.12 | 82.06 | 81.36 | 81.12 | 80.91 | 80.69 |
| ConvNova (1 dilation) | 76.90 | 77.14 | 77.28 | 77.73 | 76.80 | 76.59 | 77.24 | 77.26 | 77.68 | 76.96 |
| Caduceus-Ph | 80.29 | 79.85 | 78.63 | 81.47 | 80.24 | 81.50 | 81.35 | 80.35 | 79.95 | 81.02 |
| HyenaDNA | 76.88 | 77.01 | 78.23 | 77.24 | 79.84 | 78.39 | 77.56 | 79.27 | 79.11 | 77.90 |
| DNABERT-2 | 79.61 | 79.34 | 79.12 | 78.90 | 79.02 | 79.61 | 78.68 | 79.42 | 79.99 | 79.41 |
| NTv2 | 76.55 | 80.32 | 78.30 | 77.52 | 80.71 | 79.25 | 76.89 | 77.82 | 76.94 | 77.38 |
| **H3K4me1** | | | | | | | | | | |
| ConvNova (4 dilation) | 58.88 | 56.99 | 56.37 | 56.20 | 56.70 | 55.59 | 55.71 | 55.71 | 56.48 | 57.41 |
| ConvNova (1 dilation) | 55.67 | 57.8 | 56.62 | 57.64 | 55.28 | 55.65 | 55.80 | 56.32 | 57.44 | 55.82 |
| Caduceus-Ph | 53.31 | 53.00 | 54.54 | 51.87 | 53.40 | 53.42 | 52.34 | 51.91 | 52.01 | 52.45 |
| HyenaDNA | 43.04 | 44.54 | 44.04 | 44.55 | 46.02 | 44.9 | 41.93 | 45.26 | 46.13 | 44.77 |
| DNABERT-2 | 50.32 | 47.21 | 48.09 | 49.38 | 49.22 | 43.71 | 46.91 | 50.22 | 47.82 | 50.51 |
| NTv2 | 51.56 | 50.96 | 52.35 | 51.81 | 50.52 | 52.03 | 52.31 | 52.72 | 51.25 | 50.93 |
| **H3K4me2** | | | | | | | | | | |
| ConvNova (4 dilation) | 55.80 | 54.68 | 51.81 | 52.52 | 54.51 | 52.20 | 51.30 | 55.01 | 54.91 | 54.45 |
| ConvNova (1 dilation) | 56.53 | 59.72 | 57.26 | 58.60 | 56.13 | 56.35 | 58.72 | 56.25 | 58.67 | 56.24 |
| Caduceus-Ph | 47.96 | 47.23 | 52.47 | 50.14 | 51.35 | 50.14 | 52.18 | 50.01 | 48.72 | 48.56 |
| HyenaDNA | 40.5 | 40.02 | 42.43 | 43.32 | 41.80 | 44.07 | 43.78 | 42.39 | 44.82 | 43.65 |
| DNABERT-2 | 43.10 | 42.10 | 45.20 | 41.90 | 43.01 | 40.10 | 43.80 | 44.40 | 43.40 | 43.20 |
| NTv2 | 37.76 | 36.32 | 37.59 | 37.57 | 35.56 | 37.87 | 39.49 | 37.04 | 36.35 | 36.86 |
| **H3K4me3** | | | | | | | | | | |
| ConvNova (4 dilation) | 59.23 | 62.11 | 60.42 | 59.98 | 59.82 | 60.10 | 60.58 | 58.73 | 61.45 | 59.60 |
| ConvNova (1 dilation) | 67.13 | 66.8 | 67.97 | 67.79 | 66.45 | 67.01 | 66.75 | 68.00 | 67.42 | 66.22 |
| Caduceus-Ph | 56.93 | 56.09 | 58.64 | 54.56 | 57.95 | 57.84 | 58.54 | 56.49 | 54.14 | 56.03 |
| HyenaDNA | 47.65 | 52.13 | 50.12 | 52.10 | 52.32 | 51.43 | 49.46 | 49.72 | 47.26 | 51.90 |
| DNABERT-2 | 46.30 | 46.40 | 46.60 | 46.00 | 45.80 | 45.60 | 46.40 | 44.80 | 44.30 | 42.10 |
| NTv2 | 51.70 | 48.94 | 50.52 | 49.17 | 52.03 | 50.15 | 49.87 | 51.34 | 50.75 | 48.53 |
| **H3K9ac** | | | | | | | | | | |
| ConvNova (4 dilation) | 68.98 | 69.98 | 66.18 | 67.54 | 68.39 | 66.18 | 67.54 | 68.39 | 68.67 | 69.10 |
| ConvNova (1 dilation) | 63.85 | 65.33 | 67.12 | 65.15 | 66.24 | 63.66 | 65.38 | 66.82 | 65.17 | 66.20 |
| Caduceus-Ph | 63.78 | 62.28 | 64.20 | 65.04 | 61.06 | 60.98 | 65.24 | 63.59 | 63.55 | 62.95 |
| HyenaDNA | 57.57 | 57.51 | 58.37 | 60.25 | 58.57 | 59.78 | 59.19 | 59.31 | 57.53 | 56.88 |
| DNABERT-2 | 58.80 | 60.40 | 61.00 | 58.82 | 60.41 | 60.98 | 58.77 | 60.78 | 61.23 | 59.20 |
| NTv2 | 60.45 | 61.90 | 62.22 | 59.98 | 60.85 | 61.57 | 62.10 | 60.02 | 61.78 | 59.65 |
| **H3K14ac** | | | | | | | | | | |
| ConvNova (4 dilation) | 66.24 | 66.61 | 65.91 | 68.03 | 65.98 | 65.83 | 65.6 | 66.58 | 64.93 | 66.22 |
| ConvNova (1 dilation) | 69.76 | 73.03 | 69.26 | 72.57 | 69.95 | 69.88 | 71.03 | 69.46 | 72.23 | 69.92 |
| Caduceus-Ph | 61 | 59.36 | 62.4 | 60.7 | 62.14 | 57.9 | 62.83 | 61.31 | 59.65 | 61.08 |
| HyenaDNA | 56.08 | 57.59 | 54.5 | 54.31 | 57.31 | 55.19 | 58.26 | 57.09 | 57.95 | 58.85 |
| DNABERT-2 | 56.31 | 55.72 | 53.42 | 55.14 | 55.61 | 55.36 | 49.52 | 54.54 | 53.81 | 55.72 |
| NTv2 | 55.03 | 58.65 | 56.74 | 59.18 | 57.01 | 55.82 | 57.33 | 58.40 | 56.10 | 57.92 |
| **H3K36me3** | | | | | | | | | | |
| ConvNova (4 dilation) | 67.12 | 67.58 | 68.44 | 69.43 | 68.07 | 68.54 | 69.13 | 67.75 | 67.85 | 69.22 |
| ConvNova (1 dilation) | 69.31 | 67.01 | 66.1 | 66.43 | 67.17 | 68.31 | 66.66 | 66.24 | 66.59 | 67.01 |
| Caduceus-Ph | 62.53 | 61.53 | 61.16 | 62.3 | 59.68 | 61.42 | 61.46 | 60.33 | 60.81 | 59.93 |
| HyenaDNA | 60.27 | 60.52 | 59.8 | 59.01 | 58.88 | 59.66 | 60.73 | 60.09 | 60.98 | 59.3 |
| DNABERT-2 | 59.12 | 58.11 | 58.11 | 57.16 | 56.14 | 57.15 | 58.25 | 58.21 | 58.65 | 55.27 |
| NTv2 | 58.75 | 61.42 | 60.15 | 62.10 | 59.81 | 61.68 | 60.25 | 58.90 | 60.77 | 61.12 |
| **H3K79me3** | | | | | | | | | | |
| ConvNova (4 dilation) | 72.24 | 71.37 | 71.05 | 70.85 | 72.41 | 72.73 | 72.67 | 72.19 | 72.35 | 72.91 |
| ConvNova (1 dilation) | 70.99 | 71.83 | 69.34 | 72.57 | 71.17 | 70.66 | 71.85 | 70.23 | 71.24 | 71.67 |
| Caduceus-Ph | 68.78 | 68.28 | 67.77 | 68.02 | 65.14 | 66.89 | 66.37 | 66.49 | 66.58 | 67.42 |
| HyenaDNA | 62.6 | 65.71 | 67.33 | 66.82 | 66.97 | 65.52 | 65.23 | 67.68 | 67.63 | 66.99 |
| DNABERT-2 | 64.10 | 64.10 | 65.10 | 64.10 | 64.12 | 65.13 | 64.32 | 64.21 | 64.82 | 63.91 |
| NTv2 | 64.23 | 66.14 | 67.22 | 65.99 | 63.87 | 65.43 | 68.12 | 64.68 | 66.79 | 65.31 |

Table 18: **(Cont.) Results for all models on NT benchmark across genomic regions.**

| Task/Model | 2222 | 42 | 43 | 44 | 45 | 46 | 47 | 48 | 49 | 50 |
|---|---|---|---|---|---|---|---|---|---|---|
| **_H4_** | | | | | | | | | | |
| ConvNova (4 dilation) | 80.44 | 80.97 | 82.36 | 80.76 | 81.37 | 81.17 | 81.62 | 80.19 | 80.74 | 81.54 |
| ConvNova (1 dilation) | 77.29 | 76.05 | 77.06 | 76.83 | 78.82 | 76.29 | 76.88 | 77.42 | 76.81 | 79.06 |
| Caduceus-Ph | 80.51 | 80.02 | 80.21 | 79.94 | 81.12 | 79.72 | 79.36 | 79.82 | 80.25 | 80.33 |
| HyenaDNA | 78.33 | 77.61 | 77.60 | 77.84 | 77.85 | 77.96 | 79.73 | 78.58 | 79.24 | 76.79 |
| DNABERT-2 | 78.82 | 78.42 | 77.83 | 78.83 | 78.42 | 77.26 | 78.62 | 78.23 | 77.54 | 78.14 |
| NTv2 | 78.53 | 80.21 | 79.65 | 79.11 | 81.20 | 78.92 | 79.76 | 81.00 | 80.45 | 79.83 |
| **_H4ac_** | | | | | | | | | | |
| ConvNova (4 dilation) | 64.78 | 65.04 | 62.44 | 64.47 | 66.65 | 63.18 | 64.95 | 66.23 | 65.63 | 64.14 |
| ConvNova (1 dilation) | 64.9 | 66.89 | 66.77 | 66.9 | 65.68 | 65.32 | 66.14 | 65.77 | 66.82 | 65.77 |
| Caduceus-Ph | 58.21 | 60.68 | 60.68 | 60.69 | 58.96 | 57.55 | 59.52 | 60.47 | 60.27 | 55.59 |
| HyenaDNA | 53.77 | 53.99 | 51.30 | 57.10 | 55.75 | 54.30 | 51.32 | 55.36 | 55.34 | 53.22 |
| DNABERT-2 | 51.82 | 51.92 | 51.73 | 51.83 | 51.92 | 51.73 | 51.83 | 51.92 | 51.73 | 51.83 |
| NTv2 | 53.68 | 56.94 | 54.72 | 57.41 | 54.20 | 55.84 | 56.10 | 54.95 | 53.02 | 55.35 |

## A.6 RELATIVE OCCUPANCY OF YEAST HISTONE MARKS

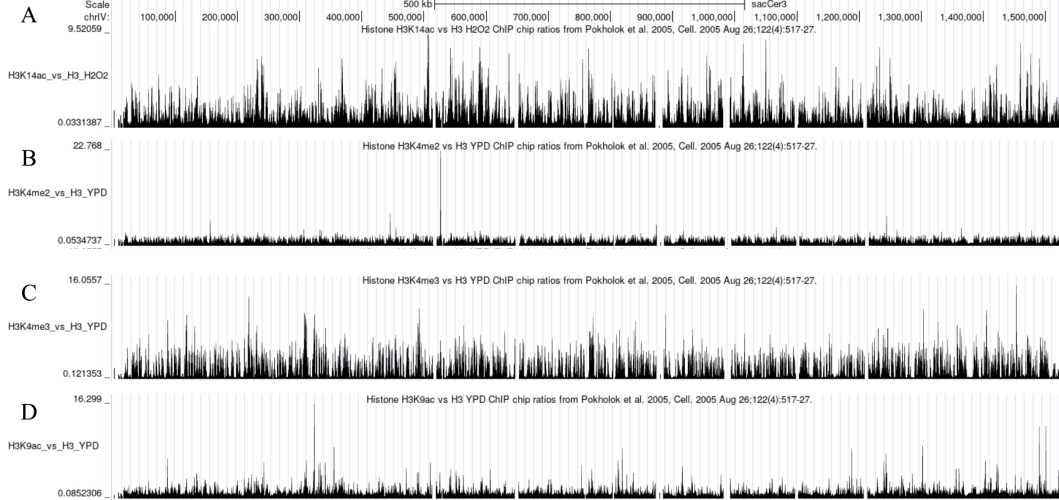

Figure 6: **Relative occupancy of histone marks on chromosome IV. A)** Histone H3K14ac vs H3 YPD ChIP chip ratios **B)** Histone H3K4me2 vs H3 YPD ChIP chip ratios **C)** Histone H3K4me3 vs H3 YPD ChIP chip ratios **D)** Histone H3K9ac vs H3 YPD ChIP chip ratios. Relative occupancy of histone marks on chromosome IV. Data were obtained from the Yeast Genome Database. For complete datasets and additional information, please refer to https://www.yeastgenome.org/dataset/E-WMIT-3#resources.

