# OpenReview forum: "Revisiting Convolution Architecture in the Realm of DNA Foundation Models"
_ICLR.cc/2025/Conference — ICLR 2025 Poster_

### Official Review · Reviewer_2bPc · 2024-10-19

**Soundness:** 3
**Presentation:** 2
**Contribution:** 2
**Rating:** 6
**Confidence:** 2

**Summary:**

The paper argues that a well-designed convolutional neural network (CNN), when used as a DNA foundation model, can outperform Transformer- and SSM-based models not only in accuracy but also in inference speed, model size, and training cost. The authors introduce ConvNova, a model composed of dual-branched Gated Convolutional Blocks (GCBs). The GCBs incorporate dilated convolutions and avoid performance-degrading downsampling, as supported by empirical findings. Ablation studies demonstrate the benefits of the dilated convolutions, dual-branch architecture, and gated mechanisms within the GCBs. The proposed model achieves higher accuracy while maintaining a smaller parameter count compared to several state-of-the-art Transformer and SSM models.

**Strengths:**

The authors conducted a series of carefully controlled experiments using different random seeds to demonstrate the accuracy advantages of their design. They compared the proposed ConvNova model against state-of-the-art Transformer- and SSM-based models across various tasks, including two benchmarks for short-range input sequences (the Nucleotide Transformer Benchmark and the Genomic Benchmark) and two benchmarks for long-range tasks (the Bend Gene Finding and the Chromatin Profile Prediction). Overall, the experiments and the ablation study are robust and effectively support the claims and the effectiveness of the proposed design.

**Weaknesses:**

While the authors present solid empirical evidence, the paper lacks a clear discussion on the intuition behind the design, making it more like a technical report. For example, ConvNeXt [1] builds on a well-known variant, ResNeXt [2], and details the accuracy impact of each modification. In contrast, ConvNova does not seem to establish a clear rationale connecting it to previous CNN-based DNA foundation models, which makes the design choices appear somewhat ad hoc. Beyond demonstrating superior accuracy compared to Transformers and SSMs, it would be valuable to include an in-depth discussion and comparison with models like LegNet on block design, architecture, parameter count, and training schemes in the main text.

I found the organization of the paper somewhat confusing. For instance, the comparison of downsampling and dilation in Section 3.3 and Table 1 would be more appropriately placed in the experiments section. Additionally, given the variety of tasks with different state-of-the-art models and settings, I recommend adding a dedicated section that briefly outlines the experimental setup. This section should include the objectives of each task, descriptions of the baseline models, and the specific configurations used in each experiment.

[1] A ConvNet for the 2020s

[2] Aggregated Residual Transformations for Deep Neural Networks

**Questions:**

- What is the intuition and rationale behind using dilation over self-attention in a DNA foundation model? And why is dilation outperforming self-attention?
- How is the receptive field for the dilated convolutions determined?
- What factors contribute to the dual-branch design outperforming a single-branch approach?
- In addition to the details in Table 11 and Section A.4, could you provide a comparison between ConvNova and LegNet in terms of structure and training scheme, and explain why your design is superior?

---

> ### Author Response · Authors · 2024-11-23
> **Response to Reviewer 2bPc(1/4)**
>
> We sincerely appreciate Reviewer 2bPc's thoughtful feedback, and here we provide corresponding responses to address these concerns.
> > While the authors present solid empirical evidence, the paper lacks a clear discussion on the intuition behind the design, making it more like a technical report. For example, ConvNeXt [1] builds on a well-known variant, ResNeXt [2], and details the accuracy impact of each modification. In contrast, ConvNova does not seem to establish a clear rationale connecting it to previous CNN-based DNA foundation models, which makes the design choices appear somewhat ad hoc.
>
> In recent foundation model research, CNNs have been largely ignored and not compared with existing methods. In fact, previous CNN models, such as Basenji, performes poorly across tasks(See Table 13). We are the first to compare CNNs with other architectures, demonstrating that with the right design, CNNs can outperform them. Indeed, this answers the question: CNNs are still not surpassed by other architectures.
>
> The motivation behind our architecture design are following several key considerations. The gating mechanism could enable dynamic feature selection. The dual-branch design facilitates independent feature extraction and promotes complementary representation learning. Dilation serves two purposes: (1) downsampling, and (2) investigating the local dependency condition of downstream tasks.
>
> However, unlike convNext, which can use ImageNet a single unified metric, DNA-related tasks are diverse and lack such benchmarks. Thus, instead of detailed analyses like ConvNeXt's, we will provide ablation studies on dilation rates and kernel sizes, offering insights into their impact across tasks soon. We will inform you once available.

---

> ### Author Response · Authors · 2024-11-23
> **Response to Reviewer 2bPc(2/4)**
>
> >Beyond demonstrating superior accuracy compared to Transformers and SSMs, it would be valuable to include an in-depth discussion and comparison with models like LegNet on block design, architecture, parameter count, and training schemes in the main text.
>
> Thank you for the suggestion. As our main focus mentioned above, we do not prioritize detailed comparisons with specific models like LegNet in the main text. However, we appreciate the importance of such comparisons and have provided the following details for reference: .
>
> **Training Schema**
> 1. LegNet's original training involved prediction of expression bin probabilities, but such tasks are absent in the foundation model benchmarks. Thus both models use original labels for supervision.
> 2. LegNet uses Lion optimizer in their original implementations, but for fair comparison, we use AdamW consistently across all models.
> 3. LegNet is trained in a fully supervised, from-scratch manner, whereas ConvNova leverages pretraining.
> 4. Even without pretraining, a from-scratch ConvNova outperforms LegNet. You can see with Table 13 and the results coming soon.
>
> **Parameter Count**
> ConvNova has multiple versions to ensure fair comparisons with different models. For comparison with LegNet, we used a 1.7M parameter version of ConvNova, while LegNet has 2.1M parameters(with the output head).
>
> **Block Design**
> 1. ConvNova incorporates dual-branch structures and dilated convolutions, which are absent in LegNet.
> 2. ConvNova’s gating mechanism is implemented using convolution, whereas LegNet uses an MLP for gating.
> 3. ConvNova maintains a fixed hidden dimension throughout, while LegNet adopts a progressively shrinking hidden dimension sequence: 256, 128, 128, 64, 64, 64, 64.
> 4. LegNet employs group convolutions to reduce parameter count, while ConvNova does not.
>
> Actually, we do not see much similarity between these two models. We hope this helps clarify the distinctions.

---

> ### Author Response · Authors · 2024-11-23
> **Response to Reviewer 2bPc(3/4)**
>
> > I found the organization of the paper somewhat confusing. For instance, the comparison of downsampling and dilation in Section 3.3 and Table 1 would be more appropriately placed in the experiments section. Additionally, given the variety of tasks with different state-of-the-art models and settings, I recommend adding a dedicated section that briefly outlines the experimental setup. This section should include the objectives of each task, descriptions of the baseline models, and the specific configurations used in each experiment.
>
> Thank you for your feedback. We agree that the comparison of downsampling and dilation would be more appropriately placed in the experiments section. We will reorganize the paper as suggested and include a dedicated section outlining the experimental setup, including the objectives of each task, descriptions of the baseline models, and the specific configurations used. We will inform you once the revisions are complete. Thank you again for your valuable input!

---

> ### Author Response · Authors · 2024-11-23
> **Response to Reviewer 2bPc(4/4)**
>
> >What is the intuition and rationale behind using dilation over self-attention in a DNA foundation model? And why is dilation outperforming self-attention?
>
> We hypothesize that DNA modeling often prioritizes interactions between adjacent nucleotides, similar to how enzymes process DNA in a sliding window fashion. This local inductive bias of CNNs, particularly with dilation, may offer an advantage over self-attention, which lacks a built-in focus on local context. This is especially evident in tasks like H3K4me2. Additionally, dilated convolutions have lower computational complexity compared to self-attention, further contributing to their efficiency and performance in DNA modeling.
>
> >How is the receptive field for the dilated convolutions determined?
>
> the receptive field can be calculated using the formula:
> $𝑅𝐹(𝑛)=𝑅𝐹(𝑛−1)+(𝑘−1)×𝑑$
> where $𝑅𝐹(𝑛)$ is the receptive field of the current layer, $𝑘$ is the kernel size, and $𝑑$ is the dilation rate.
> If your question is about how the dilation rate ($𝑑$) is determined, we select it based on ablation study results (coming soon). Additionally, since ConvNova is designed as a DNA foundation model capable of handling long-range tasks, we choose a dilation rate of 4 to balance these requirements.
>
> >What factors contribute to the dual-branch design outperforming a single-branch approach?
> The dual-branch design facilitates independent feature extraction and promotes complementary representation learning. Because of the gating mechanism, dual independent convolution can add more expressiveness than single branch.

---

> ### Author Response · Authors · 2024-11-24
> **Response to Reviewer 2bPc(with updated paper revision)**
>
> >Beyond demonstrating superior accuracy compared to Transformers and SSMs, it would be valuable to include an in-depth discussion and comparison with models like LegNet on block design, architecture, parameter count, and training schemes in the main text.
>
> We have added the comparison between LegNet and ConvNova(trained from scratch) in Table 13 to showcase the superiority.
>
> >I found the organization of the paper somewhat confusing. For instance, the comparison of downsampling and dilation in Section 3.3 and Table 1 would be more appropriately placed in the experiments section. Additionally, given the variety of tasks with different state-of-the-art models and settings, I recommend adding a dedicated section that briefly outlines the experimental setup. This section should include the objectives of each task, descriptions of the baseline models, and the specific configurations used in each experiment.
>
> We have moved the comparison of downsampling and dilation to Section 4.4, as you suggested. Additionally, we have made modifications and added further descriptions regarding the objectives, baseline models, and configurations of each task in Section A2.
>
> >How is the receptive field for the dilated convolutions determined?
>
> If your question is how we determine the dilation rate, we have included relevant experiments and details in A3.2, Tables 11 and 12.
>
> We appreciate your insights and hope these changes address your concerns. Please don't hesitate to reach out if you have any further questions.

---

> > ### Comment · Reviewer_2bPc · 2024-11-25
> >
> > I appreciate the authors' additional experiments during the discussion period. Indeed, your experimental results are solid, demonstrating superior performance across several tasks.
> >
> > However, I believe what the authors and this paper are missing is the "why"—why your ConvNet design outperforms Transformers and SSMs. Since the reasons behind this superior performance remain unclear, the proposed method appears to me to be more of a hyperparameter tuning exercise. For instance, if I understand the experiments in Tables 11 and 12 correctly, the authors performed a brute-force search on the kernel size and dilation rate.
> >
> > Additionally, should the title of the first column be "Kernel Settings" instead of "Task"?

---

> ### Author Response · Authors · 2024-11-25
>
> Thank you for your insightful comments. However, the premise of your question may be problematic. CNNs have long been established in DNA modeling [1],[2],[3], while SSMs and Transformers emerged later. Additionally, the DNA foundation model works [4],[5],[6],[7] did not compare Transformers and SSMs directly with CNN-based foundation models when introduced. Therefore, there is no evidence to suggest that Transformers and SSMs outperform CNNs.
>
> The key question in this field isn't why CNNs outperform Transformers and SSMs, but rather whether Transformers and SSMs truly surpass CNNs, and if so, why that might be the case. Given the current focus in the field on Transformer and SSM architectures, revisiting CNNs and including them in comparisons is both necessary and timely. Our paper addresses this by showing that under well-designed settings for CNNs, neither Transformers nor SSMs have surpassed CNNs in this domain.
>
> However, we can still offer some insights into why CNNs can outperform Transformers and SSMs. CNNs inherently possess inductive biases for neighborhood modeling, which may be crucial for DNA sequence tasks. Additionally, CNNs have lower computational complexity compared to Transformers and SSMs, making them more efficient.
>
> Regarding the kernel size and dilation rate experiments, these are not the central focus of our paper. They are included to ensure rigor and provide a more comprehensive evaluation.
>
> We encourage you to revisit the settings and focus of our paper, which may further clarify our contributions. Your feedback is greatly appreciated, and we welcome continued discussion on this topic!
>
> [1] Basset: learning the regulatory code of the accessible genome with deep convolutional neural networks. Genome Res. 2016
>
> [2] Sequential regulatory activity prediction across chromosomes with convolutional neural networks. Genome Res. 2018
>
> [3] Predicting effects of noncoding variants with deep learning-based sequence model. Nat Methods. 2015
>
> [4] The Nucleotide Transformer: Building and Evaluating Robust Foundation Models for Human Genomics. bioRxiv. 2023
>
> [5] DNABERT-2: Efficient Foundation Model and Benchmark For Multi-Species Genome. ICLR. 2024
>
> [6] HyenaDNA: Long-Range Genomic Sequence Modeling at Single Nucleotide Resolution. NeurIPS. 2023.
>
> [7] Caduceus: Bi-Directional Equivariant Long-Range DNA Sequence Modeling. ICML. 2024

---

> > ### Comment · Reviewer_2bPc · 2024-11-25
> >
> > Thank you for your reply. I agree that the computational complexity and model sizes of CNNs are appealing features. Additionally, the authors highlight a key observation in their response: "CNNs inherently possess inductive biases for neighborhood modeling, which may be crucial for DNA sequence tasks." If this is the case, could you at least provide empirical results to justify your argument? For example, demonstrating that the attention maps from Transformers and SSMs primarily focus on neighborhood sequences, thereby supporting the claim that CNNs are better options in terms of computational cost and performance for DNA sequence tasks.

---

> ### Author Response · Authors · 2024-11-25
>
> The main issue with self-attention is that it does not inherently focus on neighborhood sequences, which could contribute to their suboptimal performance in many DNA modeling tasks. To support our claim, we modify the NTv2 model by adjusting the RoPE's
> 𝜃 and initializing the bias of the 𝑞𝑘 linear layer to [0, 0, ..., 1], while keeping other initializations consistent with NTv2 (𝑠𝑡𝑑=0.02, 𝑚𝑒𝑎𝑛=0). This adjustment increases the attention map's focus on neighborhood sequences, enhancing the Transformer's inductive bias.
>
> Our approach is similar to the methodology in [1]. When tested on H3K14ac, a task with strong local dependencies, the results improved significantly (34.42 vs. 46.65 with the enhancements).
>
> However, this method provides only a naive way to add inductive bias to Transformers. It still does not surpass ConvNova trained from scratch. Further research is needed to fundamentally strengthen the Transformer's inductive bias for neighborhood modeling.
>
> While this is not our main focus, we still hope this addresses your concerns.
>
> [1] RepMLPNet: Hierarchical Vision MLP with Re-parameterized Locality. CVPR. 2022

---

> > ### Comment · Reviewer_2bPc · 2024-11-25
> >
> > Thanks for the reply. Would you kindly point out the tables/experiments for the modified NTv2?

---

> ### Author Response · Authors · 2024-11-26
>
> Thank you for your message. We have included the relevant details in the paper revision. Please refer to A5, Figure 5, and Table 15 for the updated information.
>
> Let us know if you need further clarification!

---

> > ### Comment · Reviewer_2bPc · 2024-11-27
> >
> > Thanks for the clarification. I believe A5, Figure 5, and Table 15 provide strong evidence to support the authors' claims. I would recommend that the authors extract some of the visualizations and results and include them in the main paper. This would help readers better understand the authors' motivations. Based on the additional visual evidence and results, I am happy to raise my score.

---

### Official Review · Reviewer_SH7n · 2024-11-03

**Soundness:** 3
**Presentation:** 3
**Contribution:** 2
**Rating:** 6
**Confidence:** 3

**Summary:**

The paper proposes to revisit CNN based architectures for DNA sequence modeling, and propose a new CNN based architecture, ConvNova. The authors show, with an extensive empirical study,  that the model has state of the art performance on several DNA modeling task when controlling for model size.

**Strengths:**

The authors have a high quality set of evaluation experiments for their method. This can be significant as it may lead to further research on CNN based architectures for sequence modeling. The writing is mostly clear and the author's proposal significantly improves the state of the art.

**Weaknesses:**

The novelty of the approach is somewhat limited, as CNNs have already been applied to this setting, and the training procedure is broadly the same as in the HyenaDNA paper.

The CNN architecture used is based on gated convolutions, which has already been proposed elsewhere, limiting the novelty.

To better understand the novelty of the approach, a benchmark comparing with a more standard CNN architecture would have been useful.

The paper seems to lack a clear motivation as to why this specific CNN architecture was proposed.

I would have expected to find a description of the pretraining data in the main paper.

I would say that calling the model a “foundation model” is somewhat misleading as the model used in the paper has 7M parameters and was pretrained for 14 hours on a single GPU.

**Questions:**

What is the pretraining dataset?

Did you benchmark the model against a more traditional CNN architecture?

Is the architecture inspired by the DNA task in some way?

---

> ### Author Response · Authors · 2024-11-23
> **Response to Reviewer SH7n(1/3)**
>
> We sincerely appreciate Reviewer SH7n's thoughtful feedback, and here we provide corresponding responses to address these concerns.
> >The novelty of the approach is somewhat limited, as CNNs have already been applied to this setting, and the training procedure is broadly the same as in the HyenaDNA paper.
>
> >The CNN architecture used is based on gated convolutions, which has already been proposed elsewhere, limiting the novelty.
>
> >To better understand the novelty of the approach, a benchmark comparing with a more standard CNN architecture would have been useful.
> >
> > The paper seems to lack a clear motivation as to why this specific CNN architecture was proposed.
>
> In recent foundation model research, CNNs have been largely ignored and not compared with existing methods. We are the **first to compare CNNs with other architectures, demonstrating that with the right design, CNNs can outperform them**. Indeed, this answers the question: **CNNs are still not surpassed by other architectures.** Thus, novel architecture is not the primary focus of our work.
>
> The motivation behind our architecture design is following several key considerations. The gating mechanism could enable dynamic feature selection. The dual-branch design facilitates independent feature extraction and promotes complementary representation learning. Dilation serves two purposes: (1) downsampling, and (2) investigating the local dependency condition of downstream tasks.
>
> Standard CNN models like Basenji and LegNet are designed for specific tasks and lack the generalizability required for DNA foundation modeling (See Table 13). This broader scope differentiates our work from prior efforts.

---

> ### Author Response · Authors · 2024-11-23
> **Response to Reviewer SH7n(2/3)**
>
> >I would have expected to find a description of the pretraining data in the main paper.
>
> Thank you for pointing this out. In our paper, we mention that our pretraining paradigm follows HyenaDNA, which implicitly includes a description of the pretraining data. However, we appreciate your suggestion and will explicitly clarify the details of the pretraining data in A.1 to ensure there is no ambiguity. We will inform you as soon as we've made the revision.

---

> ### Author Response · Authors · 2024-11-23
> **Response to Reviewer SH7n(3/3)**
>
> >I would say that calling the model a “foundation model” is somewhat misleading as the model used in the paper has 7M parameters and was pretrained for 14 hours on a single GPU.
>
> Thank you for the comment. Following prior works like HyenaDNA, DNABERT-2 and Caduceus, we adopt the term "foundation model." While our model is relatively small, we feel its evaluation across such diverse tasks justifies this terminology, though we appreciate differing interpretations.
>
> >Is the architecture inspired by the DNA task in some way?
>
> Yes. The key factor influencing robust performance is the carefully designed dilation mechanism. We will provide the ablation for dilation, kernel size as well as local dependency for tasks soon. We will inform you once available.

---

> ### Author Response · Authors · 2024-11-24
> **Response to Reviewer SH7n(with updated paper revision)**
>
> We have added a description of the pretraining data in Section 3.3 of the latest revision.
>
> We also have added the motivation why we adopt the dual-branch design in Section 3.2. The formula in Section A.3.1 has been revised in the hope that it will help you discover how the dual-branch structure with two independent convolutions provides greater expressiveness compared to a single-branch convolution.
>
> More ablation experiments are conducted in Section A.3.2, table 11 and table 12 to explain why we design the dilation rate mechanism.
>
> We hope this provides the clarity you were looking for, and please feel free to reach out if you have any further questions.

---

> ### Author Response · Authors · 2024-11-27
>
> Dear reviewer:
>
> Here we provide the motivation to consider CNNs against other models for DNA modeling tasks, **which has not been discussed in other work.**
>
> Besides the computational complexity and model sizes of CNNs, **CNNs inherently possess inductive biases for neighborhood modeling, which may be crucial for many DNA sequence tasks**. We have already added relevant experiments and analyses; please refer to A5, Figure 5, and Table 15 for the updated information.
>
> Here we briefly explain how we support our hypothesis.
>
> Specifically, the main issue with self-attention is that it does not inherently focus on neighborhood sequences, which could contribute to its suboptimal performance in many DNA modeling tasks. To support our claim, we modify the NTv2 model by adjusting the RoPE's 𝜃 and initializing the bias of the 𝑞𝑘 linear layer to [0,0,…,1], while keeping other initializations consistent with NTv2 (std=0.02,mean=0). This adjustment increases the attention map's focus on neighborhood sequences, thereby enhancing the Transformer's inductive bias.
>
> Our approach is similar to the methodology in [1]. When tested on H3K14ac, a task with strong local dependencies, the results show significant improvement (34.42 vs. 46.65 with the enhancements). Similar results on other 2 tasks can be found in Table 15.
>
> However, this method provides only a naive way to add inductive bias to Transformers. It still does not surpass ConvNova trained from scratch. Further research is needed to fundamentally strengthen the Transformer's inductive bias for neighborhood modeling.
>
> We welcome any additional questions or feedback you might have and would be glad to engage in further discussion.
>
> [1] RepMLPNet: Hierarchical Vision MLP with Re-parameterized Locality. CVPR. 2022.

---

> > ### Comment · Reviewer_SH7n · 2024-12-02
> > **Official Comment by Reviewer SH7n**
> >
> > The rebuttal have addressed my concerns. I will maintain my positive score.

---

### Official Review · Reviewer_PceR · 2024-11-03

**Soundness:** 3
**Presentation:** 2
**Contribution:** 3
**Rating:** 6
**Confidence:** 3

**Summary:**

The paper introduces a new convolutional-based DNA foundation model and shows that CNNs can be competitive with Transformers and SSMs both in terms of absolute performance as well as accuracy vs. speed tradeoffs

**Strengths:**

The paper presents strong, rigorous, and detailed results, with several strong baselines
The significance of the model itself is high, both for the application domain but what it means in the context of SSMs.
To me, DNA seems like the prime use case that SSMs, based on their performance profile, should be the winning model. A pure convolutional model outperforming Cadesus is a very important result in the context, and adds to the growing body of work that show SSMs and their complexities may not be necessary.

**Weaknesses:**

- The paper needs to do a better job, in the main text, of demonstrating what specifically about the model is novel, and what specifically contributes to the superior performance of this model compared to prior CNN-based DNA models.

For example, is it just better pretraining data? Is it including the complement sequence? I saw dilations, and receptive field choice were shown to be a portion of this, and I saw the ablations on the different components.

However, the specific novelty of this design is a bit ambiguous to me. The figures need to be clearer to emphasize the novel components compared to prior CNN-based models. Specifically, just answering the question in explicit detail: CNNs have been around for a while, why didn't we find a high-performing model like this well before this? What new has been added that wasn't found before?

There should be some more qualitative analysis in the main text, mainly I would like to see an understanding of how the different model classes compare in the types of errors they make. For example, does the smaller receptive field contribute to certain types of errors for tasks where longer range reasoning is necessary?

**Questions:**

Also mentioned in the weaknesses section, CNNs have been around for a while, why didn't we find a high-performing model like this well before this? What new has been added that wasn't found before?

---

> ### Author Response · Authors · 2024-11-23
> **Response to Reviewer PceR(1/2)**
>
> We sincerely appreciate Reviewer PceR's thoughtful feedback, and here we provide corresponding responses to address these concerns.
>
> > The paper needs to do a better job, in the main text, of demonstrating what specifically about the model is novel, and what specifically contributes to the superior performance of this model compared to prior CNN-based DNA models.
> For example, is it just better pretraining data? Is it including the complement sequence? I saw dilations, and receptive field choice were shown to be a portion of this, and I saw the ablations on the different components.
> >
> >However, the specific novelty of this design is a bit ambiguous to me. The figures need to be clearer to emphasize the novel components compared to prior CNN-based models. Specifically, just answering the question in explicit detail: CNNs have been around for a while, why didn't we find a high-performing model like this well before this? What new has been added that wasn't found before?
>
> In recent foundation model research, CNNs have been largely ignored and not compared with existing methods. In fact, previous CNN models, such as Basenji, performes poorly across tasks(See Table 13). We are the **first to compare CNNs with other architectures, demonstrating that with the right design, CNNs can outperform them**. Indeed, this answers the question: **CNNs are still not surpassed by other architectures.**
>
> Your points are insightful and highlight considerations that, while not our primary focus, remain important. Through additional ablation experiments(we will provide the results soon), we find that the **most critical factor is the appropriate dilation mechanism**, as deviations from optimal dilation rates lead to performance degradation. Other design choices, such as the dual-branch architecture, pretraining data, and gating mechanism, also contribute to performance improvements, as shown in our ablation studies Table 5.
>
> As for the question, "why didn't we find a high-performing model like this well before this?" the main reason is that this is an underexplored area. **We are the first to systematically evaluate CNNs across such a broad range of tasks**. What's more, traditional CNN designs(like DeepSEA) often rely on pooling operations, which are generally unsuitable for masked language modeling. By addressing these gaps, we demonstrate that with thoughtful design, CNNs can perform robustly as DNA foundation models.

---

> ### Author Response · Authors · 2024-11-23
> **Response to Reviewer PceR(2/2)**
>
> >There should be some more qualitative analysis in the main text, mainly I would like to see an understanding of how the different model classes compare in the types of errors they make. For example, does the smaller receptive field contribute to certain types of errors for tasks where longer range reasoning is necessary?
>
> This is an excellent question. From our experiments, we observe that sufficiently large pretrained transformers(like NTv2) excel at tasks with short sequences (a few hundred bp) but minimal local dependencies, such as splice site donor and acceptor prediction. However, transformers struggle on tasks like H3K4me2, which heavily rely on local dependencies. This may be due to the lack of an inductive bias in self-attention that emphasizes adjacent tokens, as CNNs naturally do.
>
> SSM models, on the other hand, perform well on long-range tasks, and they do not exhibit specific type of error.
>
> Traditional CNNs, such as LegNet, often underperform on targeted tasks like splice site prediction. However, with our design adjustments, **CNNs achieve robust performance across most tasks**, including long-range ones like gene finding(Table 2) and variant effect prediction:
>
> | Params |**ConvNova(25M)** | **Caduceus(8M)** | **HyenaDNA(1.6M)** | **HyenaDNA(0.4M)** | **HyenaDNA(3.3M)** | **HyenaDNA(6.6M)** | **NT(50M)** | **NT(100M)** | **NT(250M)** | **NT(500M)** |
> |---------|-----------------|---|-----------|--------------|--------------|--------------|--------|--------|--------|--------|
> |  Context Length (bp) | 10K  | 131K |  1K   | 16K  |  32K  | 160K | 12K | 12K | 12K | 12K | 196K |
> | AUC-ROC              |    **0.742** |  0.715  | 0.705    | 0.704        | 0.713        | 0.706        | 0.714  | 0.722  | 0.721  | 0.719  |
> | Accuracy             |    **0.667**  | 0.666  | 0.648    | 0.649        | 0.658        | 0.647        | 0.661  | 0.664  | 0.661  | 0.657  |
>
>
> We hope these clarifications and additional results address the concerns and demonstrate the value of our approach.

---

> ### Author Response · Authors · 2024-11-24
> **Response to Reviewer PceR(with updated paper revision)**
>
> We have provided ablation studies on dilation and kernel size in A3.2, Tables 11 and 12. For a comprehensive understanding, we recommend reviewing these tables alongside Table 5 to observe the impact of our proposed components on performance. We hope this helps clarify the details, and please feel free to reach out if you have any further questions.

---

> ### Author Response · Authors · 2024-11-27
>
> Dear reviewer:
>
> Here we provide the motivation to consider CNNs against other models for DNA modeling tasks, **which has not been discussed in other work.**
>
> Besides the computational complexity and model sizes of CNNs, **CNNs inherently possess inductive biases for neighborhood modeling, which may be crucial for many DNA sequence tasks**. We have already added relevant experiments and analyses; please refer to A5, Figure 5, and Table 15 for the updated information.
>
> Here we briefly explain how we support our hypothesis.
>
> Specifically, the main issue with self-attention is that it does not inherently focus on neighborhood sequences, which could contribute to its suboptimal performance in many DNA modeling tasks. To support our claim, we modify the NTv2 model by adjusting the RoPE's 𝜃 and initializing the bias of the 𝑞𝑘 linear layer to [0,0,…,1], while keeping other initializations consistent with NTv2 (std=0.02,mean=0). This adjustment increases the attention map's focus on neighborhood sequences, thereby enhancing the Transformer's inductive bias.
>
> Our approach is similar to the methodology in [1]. When tested on H3K14ac, a task with strong local dependencies, the results show significant improvement (34.42 vs. 46.65 with the enhancements). Similar results on other 2 tasks can be found in Table 15.
>
> However, this method provides only a naive way to add inductive bias to Transformers. It still does not surpass ConvNova trained from scratch. Further research is needed to fundamentally strengthen the Transformer's inductive bias for neighborhood modeling.
>
> We welcome any additional questions or feedback you might have and would be glad to engage in further discussion.
>
> [1] RepMLPNet: Hierarchical Vision MLP with Re-parameterized Locality. CVPR. 2022.

---

> > ### Comment · Reviewer_PceR · 2024-11-27
> > **Overall Thoughts**
> >
> > Thank you for the additional information and explanation. I do think that paper could still benefit from adding in qualitative results figures into the main paper.
> >
> > I maintain that **the paper should be accepted**, but would still benefit from some more discussion in exploring the qualitative differences between model classes a bit more in the main paper.

---

> ### Author Response · Authors · 2024-11-29
> **Qualitative Results**
>
> Thank you for your thoughtful feedback and suggestions. Based on your recommendations, we conduct additional explorations to investigate the qualitative differences in the types of errors made by different model classes. The conclusion from our analysis is that there are **no significant differences in the error sequences between the model classes**.  The results of our analysis can be found in this anonymous link: https://anonymous.4open.science/r/Error-Analysis-AB32.
>
> Specifically, we focus on the H3K14ac task, which exhibited significant performance differences among the models (70.71 vs. 60.84 vs. 57.22). We analyze the error sequences from ConvNova, Caduceus, and NTv2 using t-SNE to visualize the distribution of error points from various angles. However, **no clear qualitative differences are observed in the distributions t-SNE figures**.
>
> To further ensure rigor, we train a linear SVM on the error sequences to verify whether a hyperplane could effectively separate the errors from different models(0 for ConvNova and 1 for Caduceus). The results are shown below:
>
> |       | precision | recall | f1-score | support |
> |-------|-----------|--------|----------|---------|
> | **0** | 0.55      | 0.51   | 0.53     | 600     |
> | **1** | 0.45      | 0.48   | 0.46     | 492     |
>
> Similar results are observed in another test where 1 represents error sequences from ConvNova and 0 from NTv2, as shown below:
>
> |       | precision | recall | f1-score | support |
> |-------|-----------|--------|----------|---------|
> | **0** | 0.56      | 0.52   | 0.54     | 600     |
> | **1** | 0.46      | 0.51   | 0.49     | 443     |
>
> These results suggest that **no significant qualitative differences exist in the types of errors made by different model classes numerically**.
>
> We have also used traditional DNA sequence clustering algorithms including MMseq2, ALFATClust and MeShClust v3. However, these sequence-identity-based algorithms still do not yield other insights and conclusion.
>
> We welcome further discussion and appreciate your valuable insights. Please do not hesitate to share additional thoughts or suggestions!

---

### Official Review · Reviewer_NQxy · 2024-11-04

**Soundness:** 3
**Presentation:** 2
**Contribution:** 3
**Rating:** 6
**Confidence:** 5

**Summary:**

Since various Transformer and SSM-based models are proposed in DNA language modeling, the authors conduct empirical studies of whether CNNs are truly being surpassed by these recently proposed approaches. With analysis results, this paper develops a simple yet well-designed CNN-based method called ConvNova, which identifies and proposes three effective designs: 1) dilated convolutions, 2) gated convolutions, and 3) a dual-branch framework for gating mechanisms. Through extensive empirical experiments, we demonstrate that ConvNova significantly outperforms recent methods on more than half of the tasks across several foundation model benchmarks.

**Strengths:**

* **(S1)** The paper addresses a critical problem in DNA embedding with a novel application of classical convolution, showing significant improvements in species-aware tasks.

* **(S2)** The overall presentation is well originated and easy to follow. It is clear that the authors provide step-by-step designs to improve the attention mechanism for better performance and efficiency.

**Weaknesses:**

* **(W1)** Despite the efficient design, the proposed ConvNova is somewhat simple and lacks novelty and support. The (dilated) convolution with gating branch is not a new design (proposed by MogaNet [1] in 2022 and well studied by StarNet [2] and MambaOut [3] in 2024), especially after Mamba variants came out (i.e., the implicit long convolution with gating). The authors should discuss these background works (discussing in the related work section or comparing with them), and provide more supports of why the proposed design is specially useful in DNA applications (refer to Q1 for details).

* **(W2)** Although the authors have compared with several well-known DNA models, some recently published DNA models and pre-training works are overlooked (e.g., GPN [4], VQDNA [5], and DNABERT-S [6]). Meanwhile, there are various DNA benchmarks that should be referred to and compared (classical benchmarks like GUANinE v1.0 [7], BEND [8], and GUE [9]), especially some long-range benchmarks [10] where SSM-based models work well. From my perspective, I am still not sure or not convinced that the dilated convolution with gating aggregation could consistently outperform self-attention or SSM architectures.

### Reference

[1] MogaNet: Multi-order Gated Aggregation Network. ICLR, 2024.

[2] Rewrite the Stars. CVPR, 2024.

[3] MambaOut: Do We Really Need Mamba for Vision? arXiv, 2024.

[4] DNA language models are powerful predictors of genome-wide variant effects. PNAS, 2023.

[5] VQDNA: Unleashing the Power of Vector Quantization for Multi-Species Genomic Sequence Modeling. ICML, 2024.

[6] DNABERT-S: Learning Species-Aware DNA Embedding with Genome Foundation Models. arXiv, 2024.

[7] GUANinE v1.0: Benchmark Datasets for Genomic AI Sequence-to-Function Models. bioRxiv, 2023.

[8] BEND: Benchmarking DNA Language Models on biologically meaningful tasks. ICLR, 2024.

[9] DNABERT-2: Efficient Foundation Model and Benchmark For Multi-Species Genome. ICLR, 2024.

[10] Advancing DNA Language Models: The Genomics Long-range Benchmark. ICLR Workshop on Machine Learning for Genomics Explorations, 2024.

**Questions:**

* **(Q1)** Is there any empirical analysis or theoretical support to verify that the dilated convolution is capable and more efficient than self-attention or SSM modules on the DNA tasks? For example, the authors could visualize the reception field or analysis the learned patterns to show that the dilated convolutions with gating learn better.

* **(Q2)** Some questions about the hyper-parameters in the network. How to determine the kernel size and the dilated ratio in ConvNova? Are there any implementation details of the ablation studies and providing the corresponding configurations (in addition to Table 9)? Meanwhile, is there more ablation of different designs (like analysis in [1, 2, 3]) of the gating branches as mentioned in Eq. (1) and Appendix A.3?

---

> ### Author Response · Authors · 2024-11-23
> **Response to Reviewer NQxy(1/4)**
>
> We sincerely appreciate Reviewer NQxy's thoughtful feedback, and here we provide corresponding responses to address these concerns.
>
> ## For Weaknesses:
> > (W1) Despite the efficient design, the proposed ConvNova is somewhat simple and lacks novelty and support. The (dilated) convolution with gating branch is not a new design (proposed by MogaNet [1] in 2022 and well studied by StarNet [2] and MambaOut [3] in 2024), especially after Mamba variants came out (i.e., the implicit long convolution with gating). The authors should discuss these background works (discussing in the related work section or comparing with them), and provide more supports of why the proposed design is specially useful in DNA applications (refer to Q1 for details).
>
> In recent dna foundation model research, CNNs have been largely ignored and not compared with existing methods. Previous CNN models, such as Basenji, perform poorly across tasks. We want to clarify that our primary contribution is not architectural novelty, but being the **first to propose and compare traditional CNN-based DNA foundation model with other architectures, demonstrating that with the right design, CNNs can outperform them**. Indeed, this answers the question: **CNNs are still not surpassed by other architectures**.
>
> While the architectural design is not our main focus, we would like to address the concerns about the dilation with gating mechanism. The key factor driving the model's performance is not gating but rather the carefully chosen dilation design. We have compared U-Net-style downsampling approaches and performed ablation studies on different dilation rates, selecting the current design based on the requirements of foundation models for long-range tasks. Moreover, dilation help to investigate the down stream tasks' local dependency. For instance, larger dilation sizes (e.g., 4) improve performance on H3 (81.49 vs. 77.16), while smaller sizes (e.g., 1) are better for H3K4me3 (67.15 vs. 60.20).
>
> As for the gating mechanism, while it is a known concept, its use in ConvNova is empirically validated for effectiveness in DNA modeling(Table 5). While our approach differs in several aspects, we acknowledge the contributions of MogaNet, StarNet, and MambaOut.

---

> ### Author Response · Authors · 2024-11-23
> **Response to Reviewer NQxy(2/4)**
>
> > - (W2) Although the authors have compared with several well-known DNA models, some recently published DNA models and pre-training works are overlooked (e.g., GPN [4], VQDNA [5], and DNABERT-S [6]). Meanwhile, there are various DNA benchmarks that should be referred to and compared (classical benchmarks like GUANinE v1.0 [7], BEND [8], and GUE [9]), especially some long-range benchmarks [10] where SSM-based models work well. From my perspective, I am still not sure or not convinced that the dilated convolution with gating aggregation could consistently outperform self-attention or SSM architectures.
>
> Thank you for the valuable feedback. We coduct additional comparisons with GPN and DNABERT-2 on the GUE benchmark. As shown in the following results, ConvNova outperforms the baseline in 19 out of 28 tasks.
>
> ### Epigenetic Marks Prediction
>
> | Model       | H3     | H3K14ac   | H3K36me3  | H3K4me1   | H3K4me2   | H3K4me3   |
> |-------------|--------|-----------|-----------|-----------|-----------|-----------|
> | DNABERT-2  | 79.01  | _56.23_ | _60.92_ | 48.55     | 34.35     | 42.22     |
> | DNABERT-S  | **79.42** | 52.00    | 55.10     | 48.53     | 37.19     | 36.43     |
> | GPN         | _79.30_ | 52.38    | 60.77     | _51.66_ | _40.86_ | _45.38_ |
> | ConvNova    | 76.94  | **58.43** | **61.04** | **53.11** | **42.86** | **52.21** |
>
> ---
>
> ### Epigenetic Marks Prediction and Promoter Detection
>
> | Model       | H3K79me3 | H3K9ac   | H4    | H4ac   | all     | notata  | tata   |
> |-------------|----------|----------|-------|--------|---------|---------|--------|
> | DNABERT-2  | 62.69    | _57.60_ | 77.73 | 45.66  | 85.82   | _93.33_ | 64.14  |
> | DNABERT-S  | 63.90    | 53.90    | 80.30 | 47.75  | 86.89   | **93.37** | _64.45_ |
> | GPN         | _66.15_ | 57.01    | **81.56** | _51.66_ | **89.03** | 92.02   | 53.78  |
> | ConvNova    | **67.67** | **61.75** | _80.30_ | **54.89** | _87.45_ | 92.16   | **66.06** |
>
> ---
>
> ### Transcription Factor Prediction (Human) and Core Promoter Detection
>
> | Model       | 0       | 1       | 2       | 3       | 4       | all     | notata  | tata   |
> |-------------|---------|---------|---------|---------|---------|---------|---------|--------|
> | DNABERT-2  | 68.16   | 71.39   | 66.83   | **62.05** | 75.55   | 67.88   | 66.76   | 58.38  |
> | DNABERT-S  | **69.85** | **74.65** | 65.04   | _56.28_ | 74.94   | 67.00   | _69.26_ | _74.88_ |
> | GPN         | 66.37   | _72.31_ | _73.31_ | 49.73   | _76.21_ | _70.31_ | 68.42   | 73.26  |
> | ConvNova | _69.82_ | 71.83    | **74.52** | 50.58   | **76.65** | **71.40** | **69.86** | **78.74** |
>
> ---
>
> ### Transcription Factor Prediction (Mouse) and Virus & Splice Detection
>
> | Model       | 0       | 1       | 2       | 3       | 4       | Covid   | Reconstruct |
> |-------------|---------|---------|---------|---------|---------|---------|-------------|
> | DNABERT-2  | 57.29   | _83.85_ | 78.10   | _77.53_ | 46.19   | 70.29   | 86.19       |
> | DNABERT-S  | _58.29_ | **85.14** | 74.19   | 77.25   | **50.30** | _70.78_ | 85.43       |
> | GPN         | 57.79   | 78.87   | _82.32_ | 74.59   | 43.84   | 70.46   | _88.03_   |
> | ConvNova    | **59.40** | 81.68   | **86.05** | **81.63** | _47.12_ | **73.86** | **88.29**   |
>
> Regarding VQDNA, it is not included in the comparison as our work focuses on CNN-based models versus self-attention or SSM-inspired architectures. VQDNA uses vector quantized tokenization and the DNABERT-2 architecture, which falls outside the scope of this study. Additionally, as VQDNA lacks open-source code, future work may explore comparisons with VQ-based settings.
>
> For the long-range benchmarks mentioned, the lack of open-source code presents a limitation. However, we complete the Variant Effect Prediction task across three tasks. As shown in the results, ConvNova successfully models long-range dependencies.
>
> ### **Variant Effect Prediction**
>
> | Params |**ConvNova(25M)** | **Caduceus(8M)** | **HyenaDNA(1.6M)** | **HyenaDNA(0.4M)** | **HyenaDNA(3.3M)** | **HyenaDNA(6.6M)** | **NT(50M)** | **NT(100M)** | **NT(250M)** | **NT(500M)** |
> |---------|-----------------|---|-----------|--------------|--------------|--------------|--------|--------|--------|--------|
> |  Context Length (bp) | 10K  | 131K |  1K   | 16K  |  32K  | 160K | 12K | 12K | 12K | 12K | 196K |
> | AUC-ROC              |    **0.742** |  0.715  | 0.705    | 0.704        | 0.713        | 0.706        | 0.714  | 0.722  | 0.721  | 0.719  |
> | Accuracy             |    **0.667**  | 0.666  | 0.648    | 0.649        | 0.658        | 0.647        | 0.661  | 0.664  | 0.661  | 0.657  |

---

> ### Author Response · Authors · 2024-11-23
> **Response to Reviewer NQxy(3/4)**
>
> ## For Questions:
> >(Q1) Is there any empirical analysis or theoretical support to verify that the dilated convolution is capable and more efficient than self-attention or SSM modules on the DNA tasks? For example, the authors could visualize the reception field or analysis the learned patterns to show that the dilated convolutions with gating learn better.
>
> For efficiency, dilated convolutions possess lower computational complexity compared to self-attention and SSM modules. For capability, we have conducted extensive experiments with varying sequence lengths, and the results demonstrate the effectiveness of ConvNova across DNA tasks. As for a potential explanation, we hypothesize that, similar to how enzymes interact with DNA in a sliding window fashion, DNA modeling may prioritize adjacent nucleotides. In this sense, the inductive bias of CNNs might be more advantageous than self-attention models, which may not emphasize local context in the same way. This could be a key reason for the success of CNN-based models in this domain.
>
> While we do attempt to visualize the receptive fields and analyze the learned patterns, we do not observe any particularly insightful results. We agree that interpretability is an important aspect, and this might require a more systematic and dedicated effort.

---

> ### Author Response · Authors · 2024-11-23
> **Response to Reviewer NQxy(4/4)**
>
> >(Q2) Some questions about the hyper-parameters in the network. How to determine the kernel size and the dilated ratio in ConvNova? Are there any implementation details of the ablation studies and providing the corresponding configurations (in addition to Table 9)? Meanwhile, is there more ablation of different designs (like analysis in [1, 2, 3]) of the gating branches as mentioned in Eq. (1) and Appendix A.3?
>
> We will provide additional ablation experiments soon.

---

> ### Author Response · Authors · 2024-11-24
> **Response to Reviewer NQxy's Q2(with updated paper revision)**
>
> >(Q2) Some questions about the hyper-parameters in the network. How to determine the kernel size and the dilated ratio in ConvNova? Are there any implementation details of the ablation studies and providing the corresponding configurations (in addition to Table 9)? Meanwhile, is there more ablation of different designs (like analysis in [1, 2, 3]) of the gating branches as mentioned in Eq. (1) and Appendix A.3?
>
> We have provided ablation studies regarding dilation and kernel size in A3.2, Table 11 and Table 12. In short, increasing the kernel size and dilation generally improves performance. Although some tasks may benefit from smaller dilations, we choose a dilation rate of 4 to guarantee long sequences modeling capability for the foundation model. To reduce the number of parameters, we select a kernel size of 9.
>
> Regarding the gating and dual-branch designs, we have revised the ablation study in A3.1. The learning rate, optimizer settings and other settings for this ablation study are kept consistent with ConvNova.
>
> As mentioned above, our goal is not to propose a highly novel CNN architecture, so we did not conduct further comparisons of different designs.
>
> We hope this answers your questions!

---

> ### Comment · Reviewer_NQxy · 2024-11-26
> **Reply to Authors' Rebuttal**
>
> Thanks for the detailed response, extensive experiments, and the updated manuscript. Although this manuscript is technically oriented with some straightforward design, it really provided a new insight and useful DNA foundation model, which are verified by comprehensive benchmarks. What I am still concerned about is W1. I believe the authors should further clarify the novelty of the design model with empirical analysis and well-arranged backgrounds to enhance the contribution of rethinking the pure convolution-based architectures for long-sequence tasks with DNA. Maybe provide a table or timeline figure to illustrate further the motivation that CNNs can still be competitive for the genomic tasks. Overall, after going through comments from other reviewers, I decided to raise my score to 6 and encourage the authors to further polish the manuscript and provide more ablation experiments.

---

> > ### Author Response · Authors · 2024-11-27
> >
> > Thank you for your thoughtful feedback and for raising your score! We truly appreciate your recognition of our work and the constructive suggestions. We plan to include a table, along with further empirical analysis, in future versions of the manuscript to better illustrate the motivation and contributions of our design. Thank you again for your support!

---

### Author Response · Authors · 2024-11-29
**Global Response**

We sincerely thank all the reviewers for their thoughtful feedback, constructive suggestions, and encouraging comments. Your insights have been invaluable in helping us refine our work and strengthen the manuscript. We appreciate your recognition of the contributions and potential impact of our approach and will continue to improve the paper by incorporating your suggestions. Thank you again for your time and effort in reviewing our submission.

Here, we again highlight the key contributions of our work:

1. In recent foundation model research, CNNs have largely ignored and not compared with existing methods. We are **the first to conduct a comprehensive comparison across diverse benchmarks, demonstrating that with proper design, CNNs can outperform other architectures**. This provides a clear answer to the question: **CNNs are still not surpassed by other architectures**.

2. Previous CNN models, such as Basenji, fail to perform well on all DNA sequence tasks. In contrast, we introduce ConvNova, **the first DNA foundation model based on classical CNN architecture to achieve strong performance across most tasks**. Through extensive experiments, we highlight three critical design principles: a well-designed dilation mechanism, a dual-branch structure, and gating convolution.

3. We are also **the first to explore the relationship between receptive fields and the local dependency characteristics of DNA sequences**. Our experiments reveal that **CNNs inherently possess inductive biases for neighborhood modeling, which may be essential for many DNA sequence applications**.

---

### Meta-Review · Area_Chair_Hb7W · 2024-12-13

**Metareview:**

This paper makes a strong case for reconsidering convolutional neural networks (CNNs) as competitive alternatives to Transformers and state-space models (SSMs) in DNA foundation modeling. The authors propose ConvNova, a CNN-based architecture incorporating innovative elements such as dilated convolutions, gated convolutions, and a dual-branch framework. Through extensive empirical evaluation, the authors demonstrate that ConvNova outperforms state-of-the-art models across various benchmarks, including both short- and long-range DNA tasks. ConvNova's performance improvements are coupled with lower parameter counts and faster computation, addressing practical concerns of scalability and efficiency.

The paper's contributions are particularly noteworthy in a field increasingly dominated by attention-based models. By leveraging the inductive biases inherent in CNNs, such as neighborhood modeling, the authors effectively challenge the assumption that Transformers and SSMs are universally superior. Moreover, the paper underscores the importance of architectural simplicity and principled design, showing that well-tuned CNNs can excel in specialized domains like genomics.

**Additional Comments On Reviewer Discussion:**

There were concerns about the novelty of ConvNova’s design, insufficient comparison with prior CNN models, and limited intuition behind its architectural choices. Reviewers also requested clarification on pretraining data and qualitative analysis of errors across model classes. The authors responded comprehensively, emphasizing the paper’s focus on revisiting CNNs rather than proposing fundamentally novel architectures. They added comparisons with prior models, detailed ablation studies, and reorganized the paper for clarity. The reviewers found these updates satisfactory, acknowledging the robustness of experiments and the practical significance of the findings.

---

### Decision · Program_Chairs · 2025-01-22

Accept (Poster)